# Predicting the presence of infectious virus from PCR data: A meta-analysis of SARS-CoV-2 in non-human primates

Celine E. Snedden [1], James O. Lloyd-Smith [1,2] *

1 Department of Ecology and Evolutionary Biology, University of California Los Angeles, Los Angeles, California, United States of America, 2 Department of Computational Medicine, University of California Los Angeles, Los Angeles, California, United States of America

* jlloydsmith@ucla.edu

**Data Availability Statement:** All data and code used to produce the results and figures in this manuscript are available at: https://doi.org/10.5281/zenodo.10947025.

## Abstract

Researchers and clinicians often rely on molecular assays like PCR to identify and monitor viral infections, instead of the resource-prohibitive gold standard of viral culture. However, it remains unclear when (if ever) PCR measurements of viral load are reliable indicators of replicating or infectious virus. The recent popularity of PCR protocols targeting subgenomic RNA for SARS-CoV-2 has caused further confusion, as the relationships between subgenomic RNA and standard total RNA assays are incompletely characterized and opinions differ on which RNA type better predicts culture outcomes. Here, we explore these issues by comparing total RNA, subgenomic RNA, and viral culture results from 24 studies of SARS-CoV-2 in non-human primates (including 2167 samples from 174 individuals) using custom-developed Bayesian statistical models. On out-of-sample data, our best models predict subgenomic RNA positivity from total RNA data with 91% accuracy, and they predict culture positivity with 85% accuracy. Further analyses of individual time series indicate that many apparent prediction errors may arise from issues with assay sensitivity or sample processing, suggesting true accuracy may be higher than these estimates. Total RNA and subgenomic RNA showed equivalent performance as predictors of culture positivity. Multiple cofactors (including exposure conditions, host traits, and assay protocols) influence culture predictions, yielding insights into biological and methodological sources of variation in assay outcomes–and indicating that no single threshold value applies across study designs. We also show that our model can accurately predict when an individual is no longer infectious, illustrating the potential for future models trained on human data to guide clinical decisions on case isolation. Our work shows that meta-analysis of *in vivo* data can overcome long-standing challenges arising from limited sample sizes and can yield robust insights beyond those attainable from individual studies. Our analytical pipeline offers a framework to develop similar predictive tools in other virus-host systems, including models trained on human data, which could support laboratory analyses, medical decisions, and public health guidelines.

**Funding:** J.O.L.-S. and C.E.S were both supported by the Defense Advanced Research Projects Agency DARPA PREEMPT #D18AC00031. C.E.S was also supported by the National Institutes of Health (grant 5T32 GM008185-33) and the UCLA Office of the Vice Chancellor for Research 3R Grant. J.O.L.-S. was also supported by the National Science Foundation (DEB-1557022 and DEB-2245631) and the UCLA AIDS Institute and Charity Treks. The funders had no role in study design, data collection and analysis, decision to publish, or preparation of the manuscript. The content of the article does not necessarily reflect the position or the policy of the US government, and no official endorsement should be inferred.

**Competing interests:** The authors declare there are no competing interests.

## Author summary

Although viral culture is the gold-standard method to detect replicating and infectious virus, decisions in virology research, clinical diagnostics, and public health often must rely on faster, cheaper PCR assays that detect viral genetic material. Substantial scientific effort has focused on assessing whether PCR assays (and what kind of PCR assays) can accurately predict culture outcomes, often finding conflicting results. In our study, we address this long-standing question by developing a customized statistical approach to analyze a large database of non-human primates experimentally infected with SARS-CoV-2. We demonstrate that two common PCR protocols can predict viral culture results with similarly high accuracy, as long as interpretations account for other factors such as exposure conditions, demographics, and assay protocols. For example, we show that inoculated tissues are more likely to be culture-positive (for a given PCR result) on the first day post infection than all later days post infection or non-inoculated tissue on any day–a finding that will clarify interpretation of results in experimental studies. Beyond these biological findings, we also showed that our framework can accurately identify when an individual is no longer infectious, showing the potential for future versions (trained on human data) to offer an individualized approach to ending isolation. Overall, our work presents a standardized framework to quantitatively predict viral culture outcomes based on faster and cheaper assays, which can be readily adapted to any other pathogen-host system with relevant data. Our work also demonstrates the power of (Bayesian) meta-analysis, which will be essential for the new era of data sharing in virology.

## Introduction

Assays that detect and quantify the presence of viral genetic material are invaluable tools for clinicians, virologists, and epidemiologists, since they are used to identify infections, monitor individual infection trajectories, and track population-wide disease trends. The global reliance on quantitative reverse transcription-polymerase chain reaction (RT-qPCR) during the COVID-19 pandemic underscores its importance as a fast, sensitive, and relatively inexpensive mainstay of research and public health. Yet positive RT-qPCR results do not necessarily indicate active infection or viral shedding because these assays only target and quantify viral genomic material [1,2]. Viral culture is the gold-standard method to detect infectious virus, but it is slow, labor-intensive, and requires niche resources like permissive cells and biosafety facilities. This precludes its use as a primary diagnostic in public health crises or even in standard clinical and research practices where speed and accessibility matter. The development of alternate methods to accurately characterize infectiousness is an active priority.

 Seeking a culture-free method to identify replicating virus, many studies on SARS-CoV-2 developed alternative RT-qPCR assays based on coronavirus transcription mechanisms. Within host cells, coronaviruses transcribe not only full-length genomic RNA (gRNA) but also multiple subgenomic RNAs (sgRNA), which are a nested set of RNA segments that function as mRNA for translation of some structural and accessory proteins [3]. Standard RT-qPCR protocols [4] typically amplify both gRNA and sgRNA simultaneously (henceforth termed a total RNA assay and abbreviated to 'totRNA'). Since sgRNAs are only transcribed after cellular entry and are generally not packaged into mature virions [5], sgRNA-specific assays for SARS-CoV-2 were developed as a proxy for replicating virus [6], and they have been used in various contexts, including to distinguish between replicating virus and residual inoculum in animal challenge experiments [7,8]. Many studies have also retrospectively analyzed clinical

samples with sgRNA assays to gauge evidence of local replication [6,9–12], but reports of using sgRNA for point-of-care clinical decisions are exceptionally rare [13].

Despite the popularity of sgRNA assays, their diagnostic utility relative to totRNA or gRNA assays is debated. Based on evidence that sgRNA may degrade faster than gRNA [8], is not found in virions [5], and correlates better with viral culture results [9,10,14,15], some consider sgRNA a better indicator of recent replication and infectiousness [6,12]. Others dispute these claims based on contrary findings, including evidence of similar degradation rates between sgRNA and gRNA [16–18], the discovery of membrane-associated and nuclease-resistant sgRNA [16], and analyses showing that sgRNA does not correlate better with culture outcomes [19]. Studies finding that sgRNA quantities scale linearly with totRNA prompted further claims that sgRNA quantification offers no additional value relative to totRNA [17–19], and skeptics have argued that any improved correlation between sgRNA and culture likely reflects the assay's lower sensitivity rather than true biological signal [16–18]. Meanwhile, samples with large quantities of totRNA but undetectable sgRNA or unculturable virus are widely evident in the literature, especially in animal challenge experiments, but they go largely unexplained [8,20]. These patterns highlight the complexity of the relationships among PCR assays and viral culture, and they underscore that our understanding of their relative trajectories during infection remains incomplete. Given their foundational importance for research and potentially for healthcare, many studies have called for better methods to interpret these assays and their interrelationships [21–24].

Data limitations are central to these unresolved debates on how well PCR predicts culture and whether that varies by RNA type since the generalizability of observed patterns remains unclear. Each study's sample size is typically quite small (e.g., often less than 100 RNA-positive samples), protocols differ between studies (e.g., PCR target genes, cell lines), patient demographics vary (e.g., hospitalized patients versus routine screening of university students), and analytical methods differ (e.g., descriptive statistics, logistic or linear regressions). Further unexplained variation may depend on patients' age, sex, and comorbidities, which can affect infection outcomes [25–28] but are often unaccounted for in assay comparisons. Exposure route and dose are also unknown for clinical infections, and because the true infection time is unknown, analyses of clinical data must rely on metrics like time since symptom onset [6,17,22,29,30], for which individual heterogeneity and recall bias can introduce substantial noise. Despite considerable effort to correlate RNA presence with culture outcomes, no study yet has jointly evaluated these various cofactors to identify and quantify their effects, and thus no method exists to integrate all of this information to quantitatively predict an individual's infectiousness on a per-sample basis. Instead, public health agencies have recommended isolating until obtaining two consecutive negative tests or until ten days after an individual's first positive test, where the latter was later revised to only five days depending on symptom severity and other risk factors [31]. However, some individuals experience prolonged shedding, and many individuals cease to be infectious well before testing PCR or antigen negative [6,14,19,32]. An individualized, evidence-based method to ending isolation (i.e., a precision medicine approach) could improve these practices substantially, by alleviating personal and economic burdens imposed by unnecessarily long isolation while also reducing the number of days individuals may still be infectious after release under static guidelines.

In this study, we compiled and jointly analyzed a database of non-human primate (NHP) experiments, including 24 articles that reported per-sample measurements of at least two of the following assays: totRNA, sgRNA, and viral culture. This meta-analytic design enabled larger sample sizes and knowledge of variables that are unknowable with clinical data (i.e., exposure time, dose, and route), all for a gold-standard animal model of human disease [33]. We developed a Bayesian hurdle model to predict the results from these disparate assays and

to evaluate the effects of NHP species, demographic characteristics (age, sex), exposure conditions (dose and route), time since infection, and study protocols (sample type, target gene, cell line, culture assay) on the relationships among assay outcomes. We first applied this method to predict sgRNA results from totRNA results, which enabled us to reconstruct their relative trajectories for all included individuals. Then, we tested the ability of both PCR assays to predict viral culture results. We characterized model performance on withheld data to evaluate predictive accuracy and generalizability, and we analyzed apparent prediction errors in the context of individual time courses to diagnose possible sources of these errors. Finally, we assessed our model's ability to identify when an individual is no longer infectious, which we benchmarked against standard public health guidelines implemented for humans. With this work, we aimed to: (i) uncover the fundamental relationships among SARS-CoV-2 PCR assays and the presence of infectious virus, in the most human-relevant experimental model, (ii) provide a quantitative tool that can directly support the analysis, interpretation, and comparison of SARS-CoV-2 studies conducted in NHPs, and (iii) offer a standardized framework that future models can adapt to analyze relationships among disparate assays in other pathogen-host systems.

## Methods

### Database compilation

Following many of the PRISMA guidelines for systematic literature searches [34], we constructed a comprehensive database of SARS-CoV-2 viral load and infectious virus data from non-human primate experiments (**S1 Fig**). To be included, articles were required to: (i) experimentally infect rhesus macaques (*Macaca mulatta*), cynomolgus macaques (*Macaca fascicularis*), or African green monkeys (*Chlorocebus sabaeus*) with SARS-CoV-2 (restricted to basal strains, excluding those reported with the D614G mutation or other named variant), and (ii) report quantitative or qualitative measurements of viral load (measured by RT-qPCR) or infectious virus (measured by plaque assay or endpoint titration) from at least one biological specimen for at least one individual and at least one sample time post infection. Only individuals receiving no or placebo treatments were recorded.

Of 86 studies meeting these criteria, we used the 24 articles that reported at least two of the following assays: totRNA PCR, sgRNA PCR, or viral culture (**S1 Table** and **S1 Fig**) [7,8,20,35–55]. Raw data were used when available (published or obtained via email correspondence); otherwise, one author (CES) extracted data from published figures using the package 'digitize' [56] in R [57]. Additional details of data acquisition and standardization are described in the **S1 Methods**.

### Bayesian hurdle model framework

To compare disparate assays, we developed a Bayesian hurdle model with two components: (i) a logistic regression that predicts whether assay Y will fall above the limit of detection ($Y_{>LOD}$) based on assay X, and (ii) a linear regression that describes the quantitative relationship between X and Y when both are measurable ($Y_{value}$) (**S2 Fig**). Each component may include distinct sets of additional predictor variables ($A_i$ and $B_j$, respectively). For the linear component, we incorporated hierarchical errors such that the model estimates article-specific error distributions ($\sigma_a$) based on distributions of population average errors ($\bar{\sigma}$) and error standard deviations ($\sigma_{sd}$). This captures potential differences in experimental noise among studies and protocols. The basic form of this model is as follows, where δ and β are regression coefficients associated with the predictors noted in the subscript:

Logistic

$$Y_{>LOD} \sim Bernoulli(p)$$

$$logit(p) = \gamma + \delta_X X + \sum_i \delta_{A_i} A_i$$

Linear

$$Y_{value} \sim N(y, \sigma_a)$$

$$y = \alpha + \beta_X X + \sum_j \beta_{B_j} B_j$$

$$\sigma_a \sim N(\bar{\sigma}, \sigma_{sd})$$

We evaluated the predictive performance of multiple models with different combinations of candidate predictors, and so the $\sum \delta_{A_i} A_i$ and $\sum \beta_{B_j} B_j$ terms varied for each considered model. Categorical predictors with more than two classifications were treated as unordered index variables, while binary predictors were treated as indicator variables. For instances of unknown age or sex, we marginalized over all possibilities. Unless otherwise stated, we used a threshold of 50% for the logistic components when classifying a sample as predicted positive or negative.

We first applied this framework to predict sgRNA from totRNA results (termed the 'sgRNA model'). All totRNA-negative samples are predicted to be sgRNA-negative, by definition. We then predicted viral culture results from PCR data using a parallel framework (termed the 'culture model'), with the following minor modifications: (i) we considered models depending on totRNA, sgRNA, or both as predictors, and (ii) we restricted analyses to the logistic component, given scarcity of quantitative culture results. The model predicts all RNA-negative samples are culture negative.

## Candidate predictor selection and prior sensitivity analyses

All candidate predictors were included because of hypothesized effects on the relationships among assay results, as summarized below. We chose informative priors to rule out implausible parameter values and to reflect existing knowledge on the expected direction of individual effects (outlined in the **S1 Methods**), where appropriate. Notably, prior predictive simulations confirmed variable but reasonable *a priori* expectations for these informative priors, with substantial improvement over non-informative priors that do not reflect existing knowledge (**S13 Fig**). Parameter estimates for the best models were qualitatively similar between informative and non-informative priors (**S13 Fig**).

All considered models included totRNA, sgRNA, or both as the primary predictor(s). For all models, we considered multiple demographic factors including age class, sex, and non-human primate species, given hypothesized effects on SARS-CoV-2 infection [25–27,43,58,59]. Because exposure conditions can affect initial virion and totRNA quantities, we included inoculation dose (in log10 pfu) and day post infection as candidate predictors. For day post infection, we distinguished between inoculated tissues sampled on the first day versus all other days post infection, and non-inoculated tissues on any day post infection (see **S11 Table** for tissue-specific categorization). Because sample content and processing may vary between non-invasive (e.g., swabs) and invasive samples (e.g., whole tissues obtained at necropsy), we considered sample type as a binary predictor.

We also included predictors to account for assay-specific variation. For sgRNA models, we derived a target gene predictor based on the expected number of transcripts available for amplification during each PCR protocol, given that sgRNA abundance varies by gene [60] and totRNA assays can amplify both genomic and subgenomic RNA. We distinguished between totRNA assays that amplify most ('totRNA-high'; targeting the N gene) or few sgRNA species ('totRNA-low'; E gene) and sgRNA assays that target highly expressed ('sgRNA-high'; sgN) or less expressed sgRNA species ('sgRNA-low'; sgE, sg7), resulting in four possible protocol combinations. For culture models, we used the totRNA target gene as the predictor, except for the models including only sgRNA as the primary predictor. Since viral infectivity varies among cell lines [21,61,62] and culture sensitivity differs between endpoint dilution and plaque assays [63], we included cell line and culture assay as additional predictors for culture.

## Evaluating and comparing model performance

To find the highest performing model for each investigation, we first used a forward search to identify the model with the best performance for each possible number of predictors. We used 10-fold cross-validation to evaluate each model's predictive performance on withheld data, and for each stage we selected the predictor that most increased the expected log pointwise predictive density (ELPD) [64]. Following convention, we considered an ELPD difference of less than 4 to be small when comparing two models [64]. Of those models identified by the forward search, we selected the 'best model' as the one with fewest predictors that achieved similar or better performance compared to the 'full model' (containing all predictors) on out-of-sample (test) data for three relevant statistics: (i) ELPD, (ii) prediction accuracy (i.e., the percent of correctly classified samples for the logistic component, or the percent of samples where the observed value fell within the 50% prediction interval for the linear component), and (iii) Matthew's correlation coefficient [65] (MCC; logistic components) or median absolute error on the posterior predictive medians (MAE; linear component). Comprehensive descriptions of model evaluation and selection are provided in the **S1 Methods.**

## Accounting for lab effects

Since there are other possible sources of methodological variation among articles besides target genes, cell lines, and culture assays (e.g., RNA extraction methods, sample storage conditions), we also fit all of our best models with an additional categorical predictor to account for lab effects. To reduce the risk of overfitting, when possible, we grouped labs based on where they conducted their primate experiments to account for common elements in lab protocols (e.g., many studies that analyzed sgRNA housed their primates at BIOQUAL, Inc.). Out of all articles, we identified eight groups of labs for the sgRNA analyses and ten groups of labs for the culture analyses (**S8 Table**). We incorporated the lab effect as another linear predictor to the logit probability term for the logistic components or to the mean of the normal distribution for the linear component. The error term for the linear component remained article- (not lab-) specific. We fit each of these models with the same informative priors used in the models without lab effects, and we added non-informative priors for the lab effects.

## Analyzing isolation end times

To assess performance on clinically relevant metrics, we evaluated how well our simple and best culture models can identify when an individual is no longer infectious (i.e., no longer culture positive). We restricted these analyses to individuals with at least two samples from the respiratory tract after their first positive test from the same location and sample type. For each

individual, we estimated the end of their infectious period as the midpoint between their last true observed culture positive and their next observed culture negative (S18 Fig). When this resulted in the infectious period ending on a half day, we rounded up to the nearest day, such that all individuals are assumed to be infectious from the day of their first positive test up to (but not including) the day on which they reach their calculated midpoint.

We then determined their model-specific isolation end time as the earliest day on which the associated model predicted a second consecutive culture negative, to mirror the public health guideline about two consecutive negative test results. Unless otherwise stated, we used our standard threshold of 50% to classify samples as predicted negative or predicted positive. We excluded the individuals for which neither model predicted a second consecutive negative, resulting in 77 total trajectories for this analysis. When only one of the two models was unable to identify such a time, we conservatively assumed that, under that model, the individual would isolate until day 10 after their first positive. We benchmarked our analyses against standard guidelines developed for COVID-19 patients, where individuals are released from isolation (i.e., assumed to no longer be infectious) on days five or ten after their first positive test [31]. To compare the performance of these isolation methods, we calculated: (i) the number of days each individual spent unnecessarily isolated when they were no longer infectious ('unnecessary isolation days'), and (ii) the number of days they were still infectious while no longer isolating ('non-isolated infectious days').

## Computational methods and software

All data preparation, analysis, and plotting were completed with R version 4.2.0 [57]. Posterior sampling of the Bayesian model was performed with No-U-Turn Sampling (NUTS) via the probabilistic programming language Stan [66] using the interface CmdStanR version 0.5.2. All model fits were generated by running six replicate chains with 4000 iterations each, of which the first 2000 iterations were treated as the warmup period and the final 2000 iterations were used to generate parameter estimates. Model convergence was assessed by the sampling software using $\hat{R}$, effective sample sizes, and other diagnostic measures employed by CmdStan by default. No issues were detected.

## Results

### The compiled dataset includes 2167 samples from 174 individual non-human primates

A comprehensive literature search for studies that challenged non-human primates with SARS-CoV-2 identified 24 articles that reported per-sample measurements of at least two of the following assays: totRNA RT-qPCR, sgRNA RT-qPCR, and viral culture (S1 Fig and Tables 1 and S1). Of those, 14 articles reported totRNA and sgRNA for 116 individuals and 1194 samples, and 15 articles reported viral culture and either RNA type for 90 individuals and 1315 samples. Five articles reported results for all three assays, totaling 342 such samples.

The dataset includes various demographic groups, including both sexes, ages ranging from 1 to 22 years old, and three non-human primate species (rhesus macaque, cynomolgus macaque, African green monkey) (Tables 1 and S1). The included articles span multiple study protocols, including different target genes, cell lines, exposure conditions, sample types, and sampling times. Only studies using early SARS-CoV-2 strains (i.e., excluding those reporting the D614G mutation or named variants) were included, to minimize underlying strain-specific variation. Sampling locations include the upper and lower respiratory tracts, gastrointestinal tract, and other regions.

**Table 1. Dataset summary.** Columns stratify by assay availability, including samples with results for sgRNA and totRNA, culture and either RNA type, and any combination of two or more included assays. Entries indicate sample sizes for the corresponding cofactor, formatted as: the number of samples/individuals/articles. Doses are grouped by total plaque forming units (though they are analyzed as a continuous variable). Target gene corresponds with the totRNA assay when available, otherwise the sgRNA assay. The full article-specific data distribution is shown in **S1 Table**.

| | | sgRNA & total RNA | Culture & either RNA | All data |
|---|---|---|---|---|
| **Demographics** | **Species** | | | |
| | Rhesus macaque | 640 / 78 / 11 | 476 / 46 / 9 | 1071 / 112 / 17 |
| | Cynomolgus macaque | 371 / 28 / 3 | 412 / 21 / 5 | 601 / 37 / 6 |
| | African green monkey | 183 / 10 / 1 | 427 / 23 / 4 | 495 / 25 / 4 |
| | **Age class** | | | |
| | Juvenile | 430 / 48 / 7 | 290 / 33 / 7 | 678 / 67 / 11 |
| | Adult | 667 / 56 / 10 | 993 / 50 / 9 | 1362 / 89 / 16 |
| | Geriatric | 54 / 8 / 1 | 2 / 1 / 1 | 54 / 8 / 1 |
| | Unknown | 154 / 23 / 3 | 72 / 6 / 1 | 226 / 29 / 4 |
| | **Sex** | | | |
| | Female | 673 / 57 / 11 | 803 / 47 / 12 | 1213 / 84 / 18 |
| | Male | 367 / 36 / 9 | 440 / 37 / 10 | 728 / 61 / 16 |
| | Unknown | 43 / 4 / 1 | 30 / 6 / 1 | 73 / 10 / 2 |
| **Sampling & exposure conditions** | **Exposure dose** | | | |
| | $10^4$ - $<10^6$ | 521 / 61 / 9 | 311 / 19 / 3 | 832 / 80 / 12 |
| | $\geq 10^6$ | 673 / 55 / 7 | 1004 / 71 / 12 | 1335 / 94 / 14 |
| | **Exposure route** | | | |
| | Single | 0 / 0 / 0 | 441 / 31 / 5 | 441 / 31 / 5 |
| | Multi | 1194 / 116 / 14 | 874 / 59 / 10 | 1726 / 143 / 19 |
| | **Sample type** | | | |
| | Invasive | 311 / 45 / 6 | 229 / 36 / 8 | 432 / 65 / 10 |
| | Non-invasive | 883 / 96 / 12 | 1086 / 76 / 12 | 1735 / 146 / 21 |
| | **Sample time** | | | |
| | Inoc, 1 dpi | 136 / 72 / 11 | 89 / 36 / 8 | 187 / 94 / 17 |
| | Inoc, 2+ dpi | 724 / 99 / 13 | 595 / 72 / 12 | 1160 / 145 / 21 |
| | Non-Inoc, 1+ dpi | 334 / 54 / 7 | 631 / 72 / 13 | 820 / 106 / 16 |
| **Assay protocols** | **PCR target genes** | | | |
| | N | 814 / 86 / 11 | 824 / 54 / 9 | 1435 / 120 / 17 |
| | E | 380 / 34 / 4 | 383 / 30 / 5 | 624 / 52 / 7 |
| | S | 0 / 0 / 0 | 108 / 6 / 1 | 108 / 6 / 1 |
| | **Culture assay** | | | |
| | TCID50 | — | 856 / 53 / 10 | 856 / 53 / 10 |
| | Plaque | — | 459 / 37 / 5 | 459 / 37 / 5 |
| | **Cell line** | | | |
| | Vero E6 | — | 959 / 71 / 12 | 959 / 71 / 12 |
| | Vero E6/TMPRSS2 | — | 191 / 8 / 2 | 191 / 8 / 2 |
| | Vero 76 | — | 165 / 11 / 1 | 165 / 11 / 1 |
| | ***Total*** | 1194 / 116 / 14 | 1315 / 90 / 15 | **2167 / 174 / 24** |

## Total RNA quantity does not solely explain sgRNA and culture results

Across individuals and samples in the database, totRNA, sgRNA, and culture trajectories exhibit patterns and challenges consistent with previous reports, including unexpected instances of sgRNA negativity and culture positivity (**Figs 1A and S3–S10**). Comparing PCR results, totRNA copy numbers are larger than sgRNA copy numbers when both are detectable

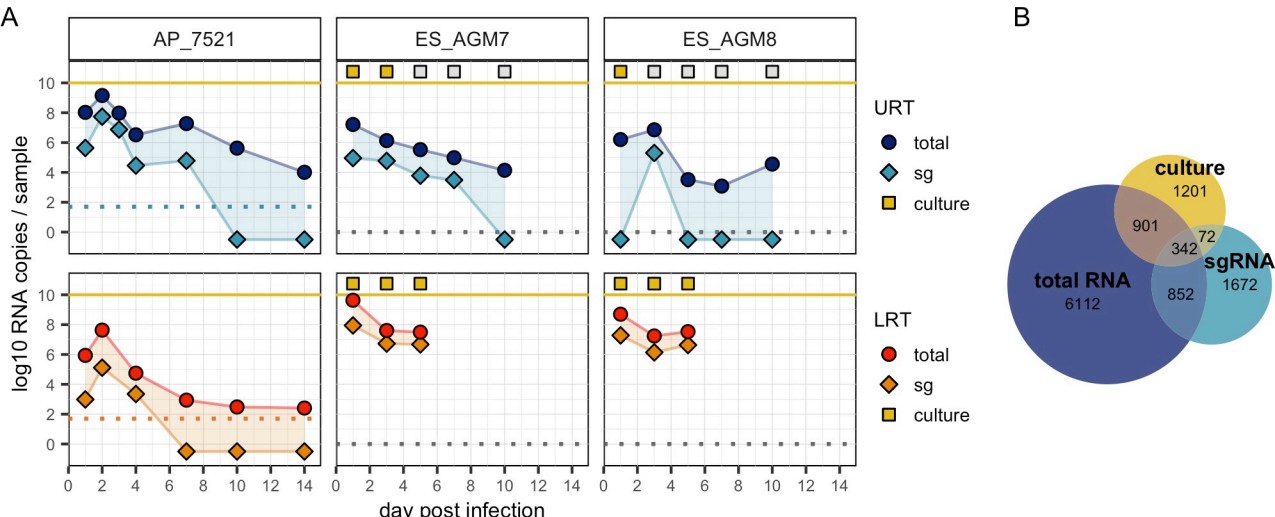

**Fig 1. Example trajectories and distribution of samples across assay types.** (A) Each column presents the totRNA (circle) and sgRNA (diamond) trajectories for the labelled individual. When available, culture results (square) are plotted above the yellow line, with yellow and grey fill indicating positive or negative culture, respectively. Samples from the upper respiratory tract (URT) are plotted above the lower respiratory tract (LRT). Dashed lines indicate reported limits of detection (plotted at 0 when unavailable). Samples with undetectable RNA are plotted below 0. Representative individuals were chosen from the full dataset. All individual trajectories are shown in **S3–S10 Figs**. (B) Number of samples available in our database for the corresponding assay(s).

(median difference: 1.45 log10 units) (**S11A Fig**), and totRNA becomes undetectable simultaneously or later in infection than sgRNA (**S11D Fig**), with rare exceptions for both patterns likely due to assay noise or processing errors. When both totRNA and sgRNA are detectable for a given individual, their trajectories are typically highly correlated (median Pearson correlation coefficient: 0.92; **S11C Fig**). However, as is particularly common in animal challenge experiments but also reported in clinical data, totRNA-positive samples in this database are often sgRNA-negative (30.0%), and totRNA quantities for these samples can be curiously large, ranging from 0.15 up to 6.38 log10 copy numbers (**S11B Fig**).

TotRNA and culture positivity results are also often discordant, disagreeing for 39.3% of all available samples and 61.3% of all totRNA-positive samples. Up to 11.02 log10 totRNA copy numbers were quantified in a culture-negative sample, which is only 1 log10 smaller than the maximum copy numbers observed in a culture-positive sample (12.09 log10) (**S11E and S11F Fig**). As few as 2.06 log10 totRNA copy numbers (when detectable) were noted in a culture-positive sample. As expected, totRNA typically becomes detectable earlier and remains detectable later than infectious virus, although for six individuals culture positivity preceded RNA positivity and one culture-positive individual was never totRNA-positive (**S11G and S11H Fig**). Considerably fewer samples with culture data also had sgRNA results (**Fig 1B**), so comparisons are limited, but patterns broadly parallel those for totRNA. Together, these patterns highlight that totRNA quantity cannot entirely explain sgRNA and culture outcomes. Statistical models may uncover cofactors underlying the discrepancies among these essential assays.

## Predictive performance on withheld data clearly identifies the best statistical models

To compare disparate assays, we developed a Bayesian hurdle model that predicts whether an assay of interest will fall above the limit of detection (the 'logistic component') and, if so, predicts a quantitative value for that assay (the 'linear component') (**S2 Fig**). We used stepwise

forward regression with 10-fold cross-validation to evaluate predictive performance on withheld data for variable numbers of predictors. This allowed us to identify the most parsimonious model with similar or better performance on three key metrics compared to the model containing all predictors (the 'best' and 'full' models, respectively). To benchmark our analysis against prior work, we also evaluated the 'simple model,' for which the logistic and linear components contain PCR results as the sole predictor (i.e., it is a hurdle model comprised of a simple logistic regression and a simple linear regression).

We first applied this method to predict sgRNA from totRNA assays (the 'sgRNA model'), for which we considered species, age class, sex, exposure dose, day post infection, PCR target gene, and sample type (invasive vs. non-invasive) as candidate predictors. We then applied the logistic model framework to relate PCR results to culture positivity (the 'culture model'), including cell line and culture assay as additional candidate predictors (see **Methods** for justifications).

For both model types, the selection procedure clearly identified the best models (**Fig 2**), where each component included a unique set of predictors. These results were robust to alternate cross-validation procedures and prior distributions. Each selected model is generalizable, as shown by comparable prediction accuracy between training and test sets. See the **S1 Methods** for further details on model evaluation and selection.

## Exposure dose, species, and PCR target gene improve predictions of sgRNA positivity

totRNA levels clearly correlate with sgRNA positivity, but the substantial overlap in totRNA quantities measured for both sgRNA-positive and sgRNA-negative samples emphasize that other factors must influence sgRNA outcomes (**Fig 3A**). The best sgRNA logistic model identified exposure dose, species, and PCR target gene as key additional predictors of sgRNA positivity (**Fig 2** and **S2 Table**). This model is highly accurate, correctly classifying 91.1% of withheld samples. It outperforms the simple model both by increasing prediction accuracy and by assigning higher probabilities to correct classifications for more samples (**Fig 3B**). For intermediate quantities of totRNA (2–6 log10 copies), sgRNA positivity predictions differ between the simple and best models (**Fig 3C**), emphasizing the particular importance of accounting for cofactors in this range. The best and full models perform similarly (**Fig 2**).

Our best model reveals insights into the three additional predictors of sgRNA outcomes: exposure dose, species, and PCR target gene. The following trends hold for model predictions across any cofactor combination when holding totRNA quantity constant: (i) individuals inoculated with larger doses have smaller chances of detecting sgRNA, (ii) African green monkeys have the smallest chance of sgRNA detection, while rhesus and cynomolgus macaques have similar predictions, and (iii) assays targeting highly-expressed sgRNA species ('sgRNA-high' assays) have higher chances of sgRNA detection than those targeting less-expressed sgRNA species ('sgRNA-low'). We refer the reader to **Fig 3C** for quantitative median predicted chances of sgRNA detection for a select cofactor combination, **Fig 3D** for qualitative variability in those predictions, and **S6 Table** for the associated 90% prediction intervals. In **S7 Table**, we also provide the 90% credible intervals for all parameters to facilitate predictions of other cofactor combinations. Columns within row groups in **Fig 3C** with a strong color gradient indicate substantial impacts of the associated cofactor on sgRNA predictions, and grey boxes highlight totRNA ranges where final classifications of sgRNA positivity differ within that cofactor group (for the standardized cofactor set).

To determine whether any of the observed patterns could stem from lab-level methodological variation, we tested whether the findings of our best model were altered by including an

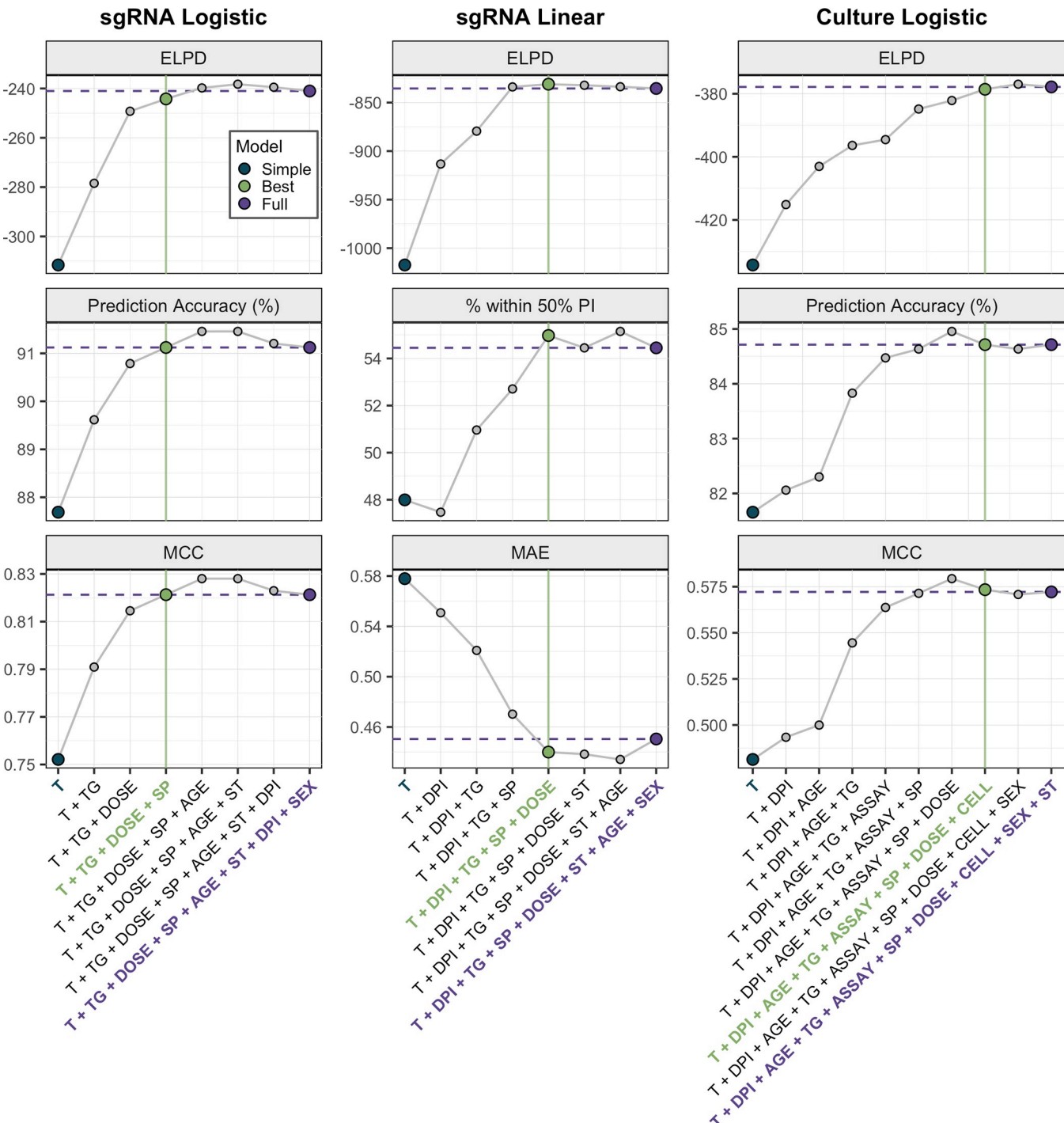

**Fig 2. Model selection criteria identify the best models.** The highest performing models for each predictor number and modeling component are shown, ordered by increasing predictor numbers. Purple horizontal lines depict performance of the full model. Green vertical lines indicate the best model, chosen according to the displayed metrics. These include estimated log pointwise predictive density (ELPD), prediction accuracy, percent of samples within the 50% prediction interval, Matthews correlation coefficient (MCC), and median absolute error around the median (MAE). These were generated using test data during 10-fold cross validation. For the culture logistic component, the model with seven predictors was not chosen because, although it outperformed the full model on MCC and prediction accuracy, it underperformed on ELPD. This is because the ELPD for the full model was larger than the ELPD for this model by more than our threshold of 4 units. Please see the **Methods** for more details about our selection criteria and the **S1 Methods** for a full description of the selection procedure. Acronyms are: T, totRNA; DPI, day post infection; SP, species; TG, target gene; ST, sample type; CELL, cell line; ASSAY, culture assay. All tested models are shown in **S2–S5 Tables**.

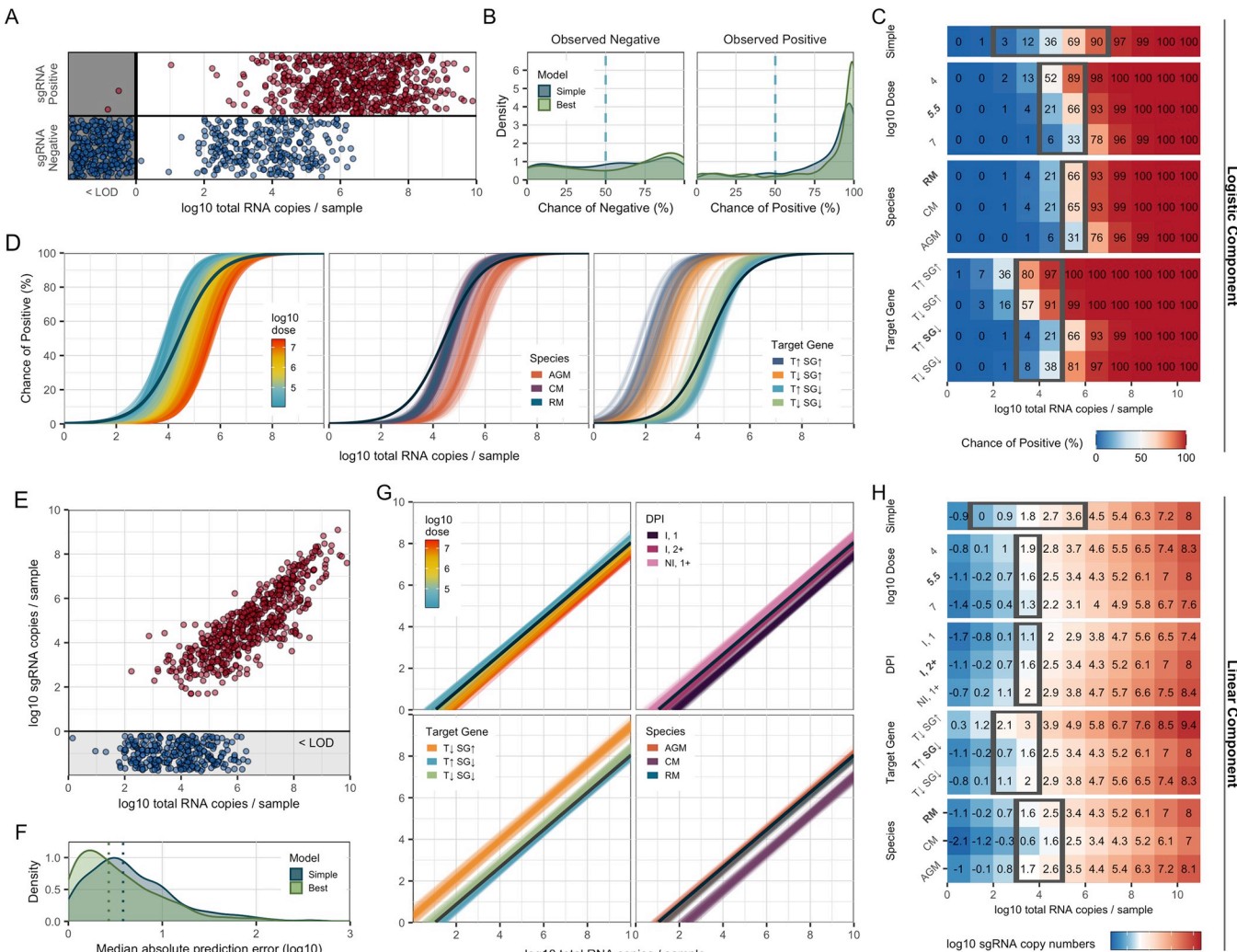

**Fig 3. The best sgRNA model captures key sources of underlying variation in PCR outcomes.** (A) All available sgRNA data plotted against totRNA results (with vertical jitter), with all totRNA-negative samples plotted in the grey region (with horizontal and vertical jitter). One totRNA- and sgRNA-positive sample with -1.18 log10 totRNA copies is not visible. (B) Distribution of median model-predicted chances of sgRNA detection for all available totRNA-positive samples, stratified by model type and observed outcomes. Samples right of the dashed line are correct predictions. (C) Median predicted chances of sgRNA detection for the simple model (top row) and all cofactor groups for the best model (other rows), evaluated for specific totRNA levels. Predictions were generated using the following 'standardized cofactor set' (which are highlighted in bold text): rhesus macaques inoculated with 5.5 log10 pfu and sampled at least two days post infection from inoculated tissues, which were processed with a totRNA-high/sgRNA-low assay. For the simple model, the grey box encloses totRNA copies where classifications differ among the simple model and any possible combination of cofactors in the best model, based on our standard prediction threshold of 50%. For all other rows, grey boxes enclose regions where classifications differ within the displayed cofactor group for the standardized cofactor set. For example, 5 log10 totRNA copies / sample is enclosed for 'Species' because African green monkeys are predicted to be negative while both other species are predicted to be positive. The rows for the other cofactor groups (e.g., target gene) do not influence the grey boxes for 'Species'. (D) 300 posterior draws from the best logistic model for the standardized cofactor set, with colored lines as indicated in panel-specific legends. The dark blue line presents the simple model's mean fit. (E) All available sgRNA data for totRNA-positive samples, where sgRNA-negative samples are plotted below 0 (with vertical jitter). (F) Distribution of median absolute errors for all sgRNA-positive samples, stratified by model type. (G) As in (D) but for the best linear component. (H) As in (C) but reporting median sgRNA copy number predictions. Grey boxes enclose regions where predicted sample quantities within the displayed cofactor group fall both above and below a common limit of detection (1.69 log10), and otherwise follow the same rules as in panel (C). Acronyms are as follows: 'RM', rhesus macaque; 'CM': cynomolgus macaque; 'AGM': African green monkey; 'Non-Inv': non-invasive; 'Inv.': invasive; 'DPI': day post infection; 'I, 1': inoculated tissues sampled on day 1 post infection; 'I, 2+': inoculated tissues sampled any other day post infection; 'NI, 1+': non-inoculated tissues on any day post infection; "T↑SG↑": totRNA-high/sgRNA-high; "T↓SG↑": totRNA-low/sgRNA-high; "T↑SG↓": totRNA-high/sgRNA-low; "T↓SG↓": totRNA-high/sgRNA-low.

additional predictor for lab effects. Some lab groups were predicted to have higher chances of sgRNA detection per totRNA quantity (**S12A Fig**), but performance was similar to the model without an explicit lab effect (**S12B Fig**). Crucially, the predicted differences among doses, species, and target genes were qualitatively unchanged between these models (**S12C and S12D Fig**), offering confidence in the robustness of our results.

## Exposure conditions, species, and PCR target gene impact expected RNA ratios

sgRNA quantities scale positively with totRNA quantities, but with considerable unexplained variation (**Fig 3E**). Our best sgRNA linear model identified exposure dose, species, PCR target gene, and day post infection as key predictors of sgRNA quantity (note these are the same predictors as for the sgRNA logistic model, but with day post infection also included). This model performs well on withheld data, with 55.0% of observed sample values falling within the model-generated 50% prediction interval (**Fig 2** and **S3 Table**). The best model clearly outperforms the simple model, decreasing the median absolute prediction error from 0.58 to 0.43 log10 copies (**Fig 3F**) and increasing the correlation between observed and median predicted values (from an adjusted $R^2$ of 0.68 to 0.77). The best model performs marginally better than the full model, with small improvements in prediction accuracy (**Fig 2**).

Below, we explore the effects of each selected cofactor on predicted sgRNA copy numbers. We report qualitative trends that hold across all cofactor combinations, and we refer the reader to **Fig 3H** for median (quantitative) predicted sgRNA copy numbers for a select cofactor combination (our 'standardized cofactor set', see figure legend). Variability in these predictions are presented qualitatively in **Fig 3G** and quantitatively (as 90% prediction intervals) in **S6 Table**. Credible intervals for all parameters are included in **S7 Table**. Similar to the logistic component, we also fit the best model with an additional predictor for lab group, which identified some modest differences in the expected sgRNA quantities among articles (**S12E Fig**) and had similar prediction accuracy to the model without lab effects (**S12F Fig**). We describe any other qualitative differences in our results between these models below, which are also visualized in **S12 Fig**.

The best model predicts that exposure conditions and sampling time impact RNA ratios. Samples obtained from individuals inoculated with larger doses must have higher total RNA copy numbers to expect the same sgRNA quantity. Results for day post infection parallel these exposure-dependent patterns. To expect a given sgRNA quantity, totRNA copies must be highest for inoculated tissues on the first day post infection, intermediate for inoculated tissues on all later days post infection, and lowest for non-inoculated tissues on any day post infection. When we added a predictor for lab group, the effects of day post infection were qualitatively unchanged while the dose effect weakened and reversed (**S12G and S12H Fig**), although a substantial portion of the parameter density allowed for the original dose effect.

PCR target genes also affect predictions. Conditional on totRNA quantity, totRNA-low/sgRNA-high assays have the largest predicted median sgRNA quantities, followed by totRNA-low/sgRNA-low and totRNA-high/sgRNA-low assays. Quantitative sgRNA outcomes were unavailable for totRNA-high/sgRNA-high assays, so estimates were not possible for those protocols. These effects were qualitatively similar in our model with lab effects (**S12G and S12H Fig**).

The best model also predicted that sgRNA quantities vary by species. Regardless of whether a lab effect was included, rhesus macaques and African green monkeys had highly similar predictions. Cynomolgus macaques were predicted to have lower median sgRNA quantities for any given totRNA quantity, though this effect was substantially reduced when lab effects were

included. Given that only one lab group had data from both cynomolgus macaques and another species (rhesus macaques), we view this species effect as an intriguing but tentative finding that warrants further investigation.

## The sgRNA model accurately reconstructs individual viral load trajectories

To further analyze performance, we reconstructed individual viral load trajectories using the best sgRNA model (**Figs 4 and S3–S5**). The model correctly predicted the timing of the first and last observed sgRNA positive for 90.1% (n = 219/243) and 72.8% (n = 177/243) of all individual- and (non-invasive) sample-specific trajectories with at least two sampling times, respectively (**S14 Fig**). Notably, 70.0% (n = 170/243) of those trajectories were predicted without a single misclassification. The distribution of predicted sgRNA quantities was highly similar to the distribution of observed sgRNA quantities (median differences of estimated means: -0.04 log10 units; 90% Credible Interval [CI]: -0.18, 0.08; **S1 Methods**) but highly dissimilar to observed totRNA values (-0.79; 90%CrI: -0.92, -0.66), offering further confidence in the model's excellent performance.

## Total RNA and sgRNA are both suitable predictors of viral culture

To determine which PCR assay best predicts viral culture, we compared models including totRNA, sgRNA, or both as predictors of culture positivity. We first evaluated performance only on samples with quantitative results for both assays and for models with no additional cofactors, for which totRNA, sgRNA, and both had similar prediction accuracy (**S9 Table**). Because few samples had both sgRNA and culture outcomes (**Fig 1B**), we imputed median sgRNA predictions where needed, using the best performing sgRNA model. On this full dataset, all three models also performed similarly well, though totRNA showed some evidence of better predicting culture positive samples. We then ran our model selection procedure on totRNA and sgRNA separately for all available data, which resulted in highly similar prediction

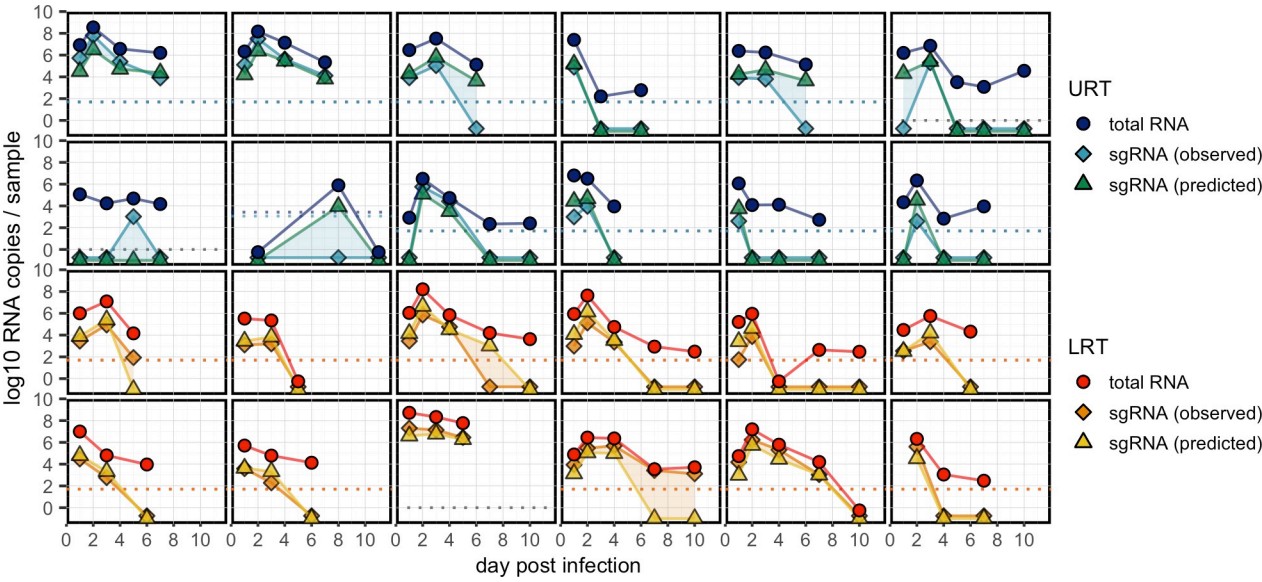

**Fig 4. The best sgRNA model reconstructs individual trajectories with high accuracy.** Each panel includes the data for one randomly selected individual sampled from either the upper respiratory (URT) or lower respiratory tract (LRT), including observed totRNA (circle), observed sgRNA (diamond), and median predicted sgRNA (triangle) quantities. Detection limits are plotted as dashed lines in the corresponding color when available, otherwise grey lines are plotted at zero. All undetectable samples are plotted below zero. See **S3–S5 Figs** for all individuals.

accuracy for both best models, though the model using totRNA was more parsimonious, with two fewer predictors (**S4 and S5 Tables**). Given this parsimony and the lack of reliance on imputed sgRNA values, plus the lack of evidence that sgRNA is a superior predictor, we based further analyses solely on totRNA.

## Demography, exposure conditions, and assay protocols resolve disparities in culture results

We next sought to predict culture positivity from totRNA results using the logistic model framework. The best model contained day post infection, inoculation dose, age class, species, culture assay, cell line, and PCR target gene as predictors, and it correctly classifies 84.7% of withheld data (**Fig 2**, and **S4 and S9 Tables**). It outperforms the simple model by correctly predicting an additional 7.0% of culture positive samples and by assigning higher probabilities for true classifications (**Figs 5B and S15A**). The difference in performance is especially pronounced at intermediate totRNA quantities (6–8 log10), which often occur during the critical transition between culture positive and negative states (i.e., in clinical terms, at the end of the infectious period). For these samples, the best model correctly predicts an additional 23.3% of culture positives (**S15B Fig**) and often with much higher confidence (**S15C Fig**). Strikingly, culture predictions can differ between the simple and best models for all considered quantities of totRNA (0–12 log10 copies) (**Fig 5C**), highlighting the benefit of accounting for cofactors when predicting culture outcomes across all totRNA quantities. The best model performs similarly to the full model (**Fig 2 and S4 Table**).

In the text below, we explore the effects of each selected cofactor on culture outcomes. Given the high dimensionality of these predictions, we report qualitative trends that hold across cofactor combinations, and we refer the reader to **Fig 5C** for median predicted chances of positive culture for a select combination of cofactors (i.e., our 'standardized cofactor set', see figure legend). Columns in **Fig 5C** with a strong color gradient indicate dramatic impacts of the associated cofactor on culture predictions, and grey boxes highlight totRNA ranges where final classifications differ within that cofactor group (for the standardized cofactor set). These ranges differ for other cofactor combinations. We present the variability of our results (for the standardized cofactor set) qualitatively in **Fig 5D** and quantitatively (as 90% prediction intervals) in **S10 Table**. In **S7 Table**, we provide the medians and 90% credible intervals for all parameters to facilitate predictions of other cofactor combinations.

To determine whether unmodelled differences among labs could explain any of the observed patterns, we fit our best culture model with an additional term for lab effects. Some groups of labs were predicted to have higher overall chances of culture positivity per totRNA quantity (**S16A Fig**), but overall prediction accuracy was similar to the model without a lab effect (**S16B Fig**). There was some additional variation in the parameter estimates for the model with a lab effect, but the qualitative findings for all cofactors were consistent across both models (**S16C and S16D Fig**).

Exposure conditions had substantial impacts on culture predictions. Individuals inoculated with larger doses have smaller probabilities of obtaining successful culture for any given totRNA quantity. Interestingly, in contrast with results predicting lower sgRNA (per totRNA quantity) in inoculated tissues (**Fig 3G and 3H**), the culture model predicts that inoculated tissues sampled on the first day post infection have the highest probabilities of being culture positive per totRNA quantity. Inoculated tissues on later days post infection and all non-inoculated tissues are much less likely to be culture positive, with substantial overlap in the predicted probabilities of those two groups.

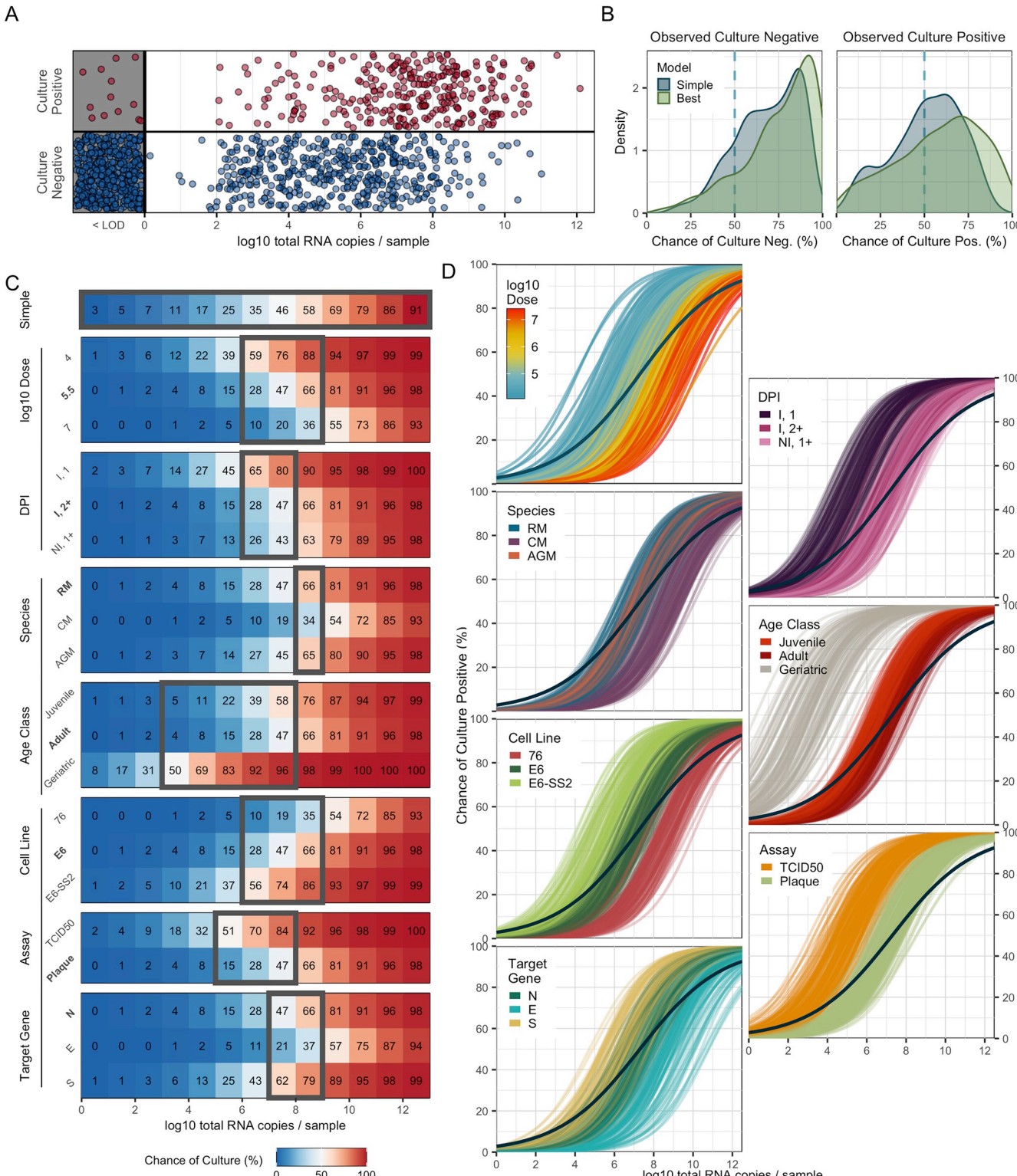

**Fig 5. The best culture model captures key sources of underlying variation in culture outcomes.** (A) All available culture data plotted against totRNA results (with vertical jitter), with all totRNA-negative samples plotted in the grey region (with horizontal and vertical jitter). (B) Distribution of median model-predicted chances of positive culture for all totRNA-positive samples, stratified by model type and observed outcomes. Samples right of the dashed vertical line are correct predictions. (C) Median predicted chance of positive culture for the simple model (top row) and all cofactor groups included in the best model (other rows) for totRNA copies (evaluated at integer values, starting at 0). Predictions were generated using the following 'standardized cofactor

set' (which are highlighted in bold text): adult rhesus macaques inoculated with 5.5 log10 pfu and sampled at least two days post infection from inoculated tissues, where PCR targets the Nucleocapsid gene and culture uses plaque assays with VeroE6 cells. Grey boxes enclose regions where classifications differ within the cofactor group for the standardized cofactor set, as described for Fig 3C. For the simple model, it encloses regions where classifications differ between the simple model and any possible combination of cofactors. (D) 300 posterior draws from the best model for the standardized cofactor set, with colored lines as indicated in panel-specific legends. The dark blue line presents the simple model's mean fit. Acronyms are as described in Fig 3, plus the following: E6, VeroE6; E6-SS2, VeroE6-TMPRSS2; and 76, Vero76 cells.

Multiple demographic factors also affect culture outcomes. Predictions for juvenile and adult age classes largely overlap, but geriatric individuals have substantially higher predicted chances of successful culture for the same viral load. This difference was reduced, though still clearly apparent, when including a lab effect. However, few samples from geriatric individuals were available (**Table 1**), and so these results should be interpreted cautiously. Predictions also vary based on species: the chances of successful culture for some viral load are smallest for cynomolgus macaques compared to rhesus macaques and African green monkeys, where the latter two species have highly similar predictions.

Assay conditions also influence culture outcomes, as expected. The model predicts that VeroE6-TMPRSS2 cells have the highest chance of positive culture, followed by VeroE6 and Vero76 cells. TCID50 assays are predicted to have higher sensitivity than plaque assays, and the chances of culture positivity (for a given viral load) are higher for PCR protocols targeting Spike (S) than for those targeting the Nucleocapsid (N) or Envelope (E) genes.

## Individual trajectories uncover sources of culture prediction errors

Although our best culture model exhibits remarkable 84.7% accuracy on withheld data, we analyzed our predictions further to identify potential causes and implications of existing errors. 64.1% (n = 116/181) of all incorrect predictions were false negatives, of which a curious 11.2% (n = 13/116) were PCR negative. Even excluding these totRNA-negative samples, totRNA copies for false negative samples were substantially smaller than for true positives (median difference of estimated population means: -2.83 log10 units; 90%CrI: -3.13, -2.53) but more similar to true negatives (median difference: 0.57; 90%CrI: 0.27, 0.87). These RNA-low but culture-positive samples could be explained by PCR or sample processing issues resulting in the amplification of less RNA (e.g., sample degradation), or by culture contamination. Similarly, totRNA copy numbers for false positive predictions were substantially larger than for true negatives (median difference: 3.05; 90%CrI: 2.74, 3.36) but were similar to true positives (median difference: -0.35, 90%CrI: -0.66, -0.04). Culture insensitivity could explain these RNA-high but culture-negative samples.

We further characterized errors by analyzing performance in the context of individual trajectories for (non-invasive) samples with at least two sampling times (**Figs 6 and S7–S9 and S17**). Overall, the best model correctly predicted 58.3% (n = 120/206) of these trajectories without a single culture misclassification, compared to only 47.6% (n = 98/206) by the simple model. Within all trajectories, the best model made a total of 131 errors in predicting culture status of individual samples, while the simple model made 171 errors. We categorized these errors into four types: (i) samples obtained on the first or last sampling day (termed an 'edge'), (ii) samples obtained as culture results transition between positive and negative states ('transition'), (iii) samples where observed culture results change for one sampling time despite surrounding instances of the opposite classification ('data blip'), and (iv) samples where culture predictions change for one sampling time despite surrounding instances of the opposite classification ('prediction blip'). Notably, while edge errors are difficult to analyze, given limited information from surrounding time points, transitions may reflect sample quality and assay

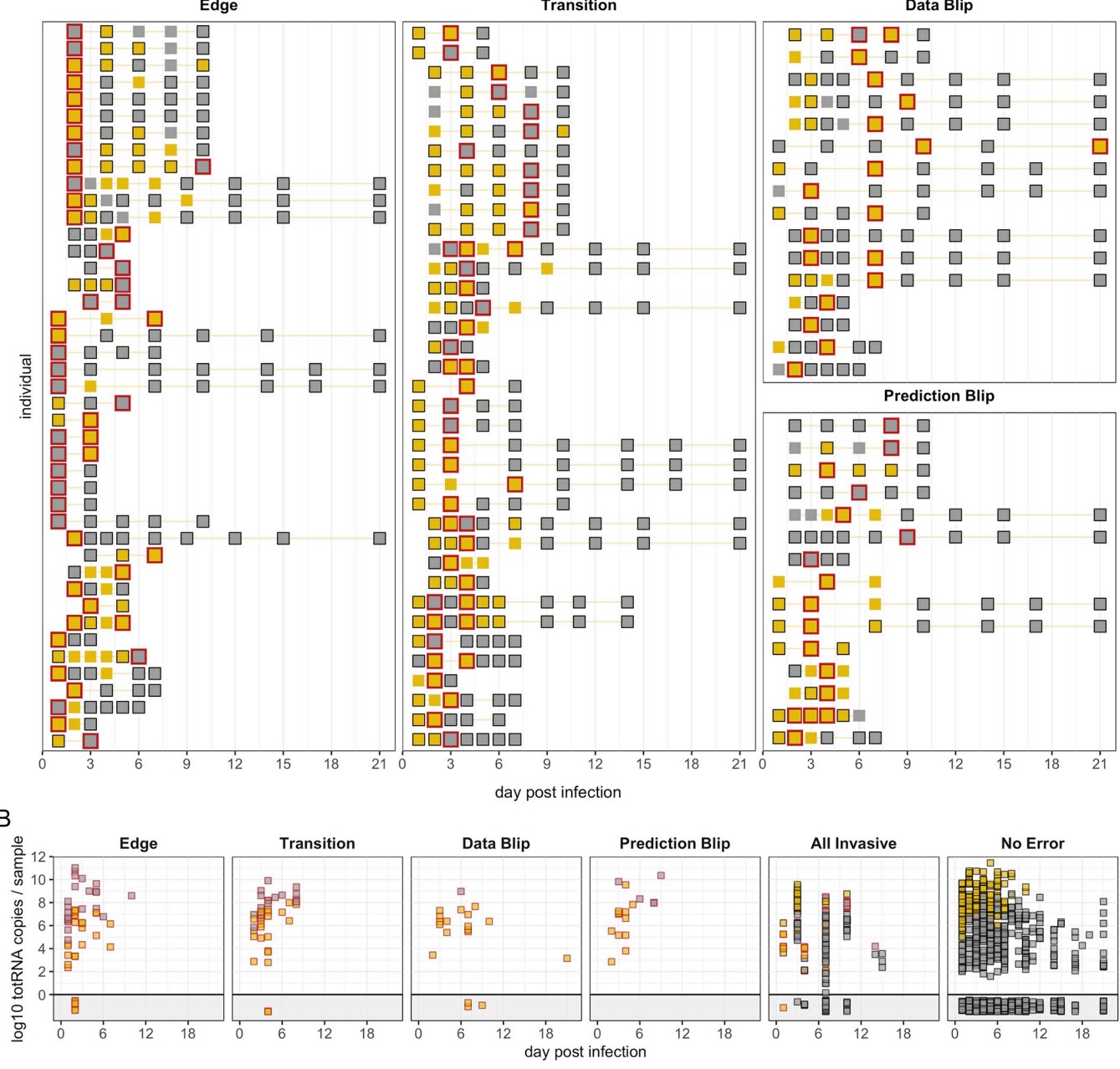

**Fig 6. Error analysis reveals potential causes of culture prediction errors.** (A) Each row shows culture results for one individual-sample trajectory that contains at least one instance of the panel-specific error type. Trajectories may appear in multiple panels if they contain multiple error types, though trajectory ordering is inconsistent. Red outlines highlight samples with the denoted error type. (B) TotRNA values over time for each error type, all invasive samples, and all correctly classified non-invasive samples ('no error'). In (A) and (B), yellow squares indicate culture positives and grey indicates culture negatives. Squares with black outline are correctly classified, while those with no or red outline are incorrectly classified. The data blip individual on day 21 post infection has another sample available at a later timepoint, so it is not considered an 'edge'.

sensitivity interacting to drive noisy outcomes for samples with intermediate RNA or virion quantities.

When considering all prediction errors, we find that edge errors are the most common for both the best (n = 51/131; 38.9%) and simple (n = 78/171; 45.6%) models. Transition errors, however, are of particular interest, given that the shift from positive to negative states determines the end of infectivity. The best model made 44 transition errors (n = 44/131; 33.6%), while the simple model made 49 transition errors (n = 49/171; 28.7%). We then calculated how many edge errors could also be considered transition errors, and once again we found that the best model made fewer such errors (23 vs. 34). Thus, model accuracy at this critical point during infection is improved by accounting for key covariates.

For the best model, data blips are less common (n = 19/131; 14.5%) than edge and transition errors, and all except one data blip are observed culture positives surrounded by culture negatives (leading to false-negative prediction errors) (**Figs 6B** and **S17A**). Eight of these samples co-occur with increases in totRNA quantities from the previous sampling time, suggesting they may reflect true local replication (e.g., as in rebound cases). The remaining instances accompany decreases in totRNA quantities, where sample contamination could drive spurious culture positivity or PCR processing issues could result in RNA underestimates. Prediction blips are the least common (n = 17/131; 13.0%), of which 70.6% (n = 12/17) are false negatives that often have lower totRNA quantities than the previous sampling time (**Figs 6B** and **S17B**). These could be explained by sample quality or PCR processing issues resulting in RNA underestimates, which is particularly plausible for instances where totRNA levels increase in the next sampling time. In contrast, false positive prediction blips primarily occur after sharp increases and high quantities of totRNA, and all occur for plaque assays. Given our model predicts lower sensitivity for plaque assays, these errors could reflect failed culture, though RNA overestimates could also explain this pattern.

## The best culture model shows potential for accurate, individualized isolation practices

Although our model is trained on NHP data and cannot be applied directly to humans, we sought to illustrate the potential clinical utility of such a framework. To do so, we assessed the simple and best models' ability to identify when an individual is no longer infectious (i.e., no longer culture positive). For all available individuals (n = 77), we determined their (model-specific) isolation end times as the earliest day on which the associated model predicted a second consecutive culture negative (**S18 Fig**). Because the time between consecutive tests increases over the course of infection (**S19 Fig**), there is an implicit bias towards longer isolation times for individuals that test positive longer and hence are observed less frequently during the period that they lose infectiousness. To account for this bias, we also ran analyses for a hypothetical 'perfect' model that identifies culture status correctly for every sample, and so it always releases individuals from isolation on the day of their true second consecutive culture negative. For further comparison, we included two standard public health guidelines for SARS-CoV-2, which release all individuals from isolation on days five or ten after their first positive test [31].

We found that, across all procedures, the best model resulted in the smallest number of days that individuals were unnecessarily isolating while no longer infectious (**Fig 7A**), with an especially large reduction compared to the ten-day protocol (126 vs. 510 days). We then considered the number of days on which individuals were not isolating but still infectious. If no isolation practices were used, there would be 260 such days. No individual was infectious up to day ten after the first positive test, and so the ten-day protocol was the only one with zero non-isolated infectious days (**Fig 7B**). The simple model had the largest number of non-isolated

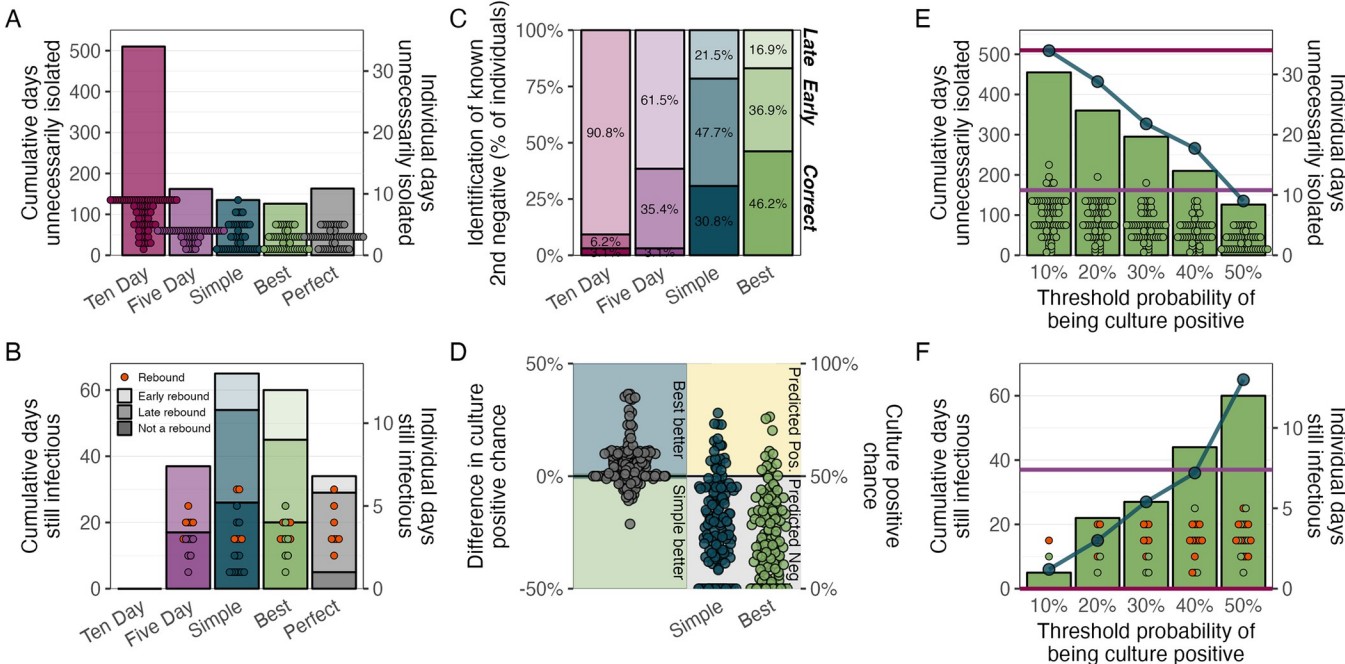

**Fig 7. The best culture model captures the end of infectiousness better than existing approaches.** (A) The cumulative days unnecessarily isolated by all individuals (histogram, left axis) and the distribution of individual days unnecessarily isolated (points, right axis) for the ten-day, five-day, simple, best, and perfect protocols. Individuals that were isolated for too few or the exact number of days are not shown. (B) The cumulative days that individuals were still infectious after the end of isolation (histogram, left axis) and the distribution of days that individuals were still infectious (points, right axis) for all the protocols in panel A. Transparency shows the classification of individual trajectories as either showing no indication of a rebound (darkest), indication of a late rebound (medium, day 5 after the first positive test or later), or indication of an early rebound (lightest, before day 5 after the first positive test). Rebound individuals are indicated by red points. Individuals that were not still infectious are not displayed. (C) Performance of each protocol on identifying the true (observed) time of the second consecutive culture negative for all individuals where this occurred. 'Correct' (darkest, bottom) includes all individuals for which the protocol exactly identified the second consecutive negative. 'Early' (medium, middle) includes all individuals where the prediction occurred before the true time, while 'Late' (lightest, top) includes all individuals where the prediction occurred after the true time. The perfect model is not shown, as by definition it is 100% correct. (D) Comparison of the culture positive probabilities predicted for the simple and best models on both samples from the first true instance of consecutive negatives. The right panel shows the raw predicted probabilities for each model. The left panel shows the per-sample difference between those probabilities for the simple and the best model, where the best model is more confident in the upper region (i.e., it has smaller predicted probabilities of being culture positive) and the simple model is more confident in the lower region. (E) The cumulative days unnecessarily isolated by all individuals (green histogram, left axis) and the distribution of individual days unnecessarily isolated (green points, right axis) for five different threshold probabilities at which a sample is considered culture positive. The best model results are displayed in the green bars (cumulative) and by the green points (individuals). The horizontal lines show the results for the five- and ten-day procedures, with the same colors as in (A). The blue points and connecting lines show the cumulative days for the simple model. (F) As in panel E, except displaying the number of days individuals were still infectious after the end of isolation. Red points are rebound individuals.

infectious days (65 days), followed by the best model (60 days), the five-day procedure (37 days), and the perfect model (34 days). Upon further investigation, many of these non-isolated infectious days arose from 16 individuals that showed evidence of rebound infection, which we defined as at least one known culture negative occurring between two known culture positives (**S18 Fig**). Of these 16 individuals, many of them (n = 6/16; 37.5%) had their final culture positive before day 5 ("early rebound"), which thus did not affect the performance of the five-day protocol but did penalize the best and simple models despite them accurately identifying many intermittent culture negatives. All protocols (except for the ten day procedure) were also affected by the 10 individuals that had their final culture positive on or after day 5 ("late rebound"; n = 10/16; 62.5%). When we excluded any rebound individuals, the best model and the five-day procedure differed by only three non-isolated infectious day (20 vs. 17 days).

To further compare the protocols, we also evaluated their ability to identify the first time that individuals experienced a true (observed) second consecutive culture negative. For these

analyses, we excluded the 12 individuals where this never occurred. We classified the protocols based on whether they accurately identified this time ('Correct') or whether the predicted time occurred before ('Early') or after ('Late') the known time. The best model was correct for the most individuals (n = 30/65; 46.2%; **Figs 7E and S18**), with the exception of the perfect model that by definition classifies all individuals correctly. The simple model only classified 30.8% (n = 20/65) of individuals correctly, which is 15.4% (n = 10/65) fewer individuals than the best model. The best model also generated 10.8% (n = 7/65) fewer early predictions, which is a particularly important improvement given the public health cost of premature release from isolation. We also analyzed the confidence with which the two models identified the first two consecutive true negatives. The best model misclassified fewer of these samples as culture positive (difference: n = 7/130; 5.4%), and it was equally or more confident (by up to 36.4%) in the correct prediction for 80.0% of the samples (n = 104/130; **Fig 7D**).

Finally, we investigated the sensitivity of our results to the threshold probability at which samples are predicted to be culture positive. We sequentially decreased this probability from 50% (our standard threshold) to a more conservative 10%, which increased the number of samples predicted to be culture positive. Because the five- and ten-day protocols are discrete rules, varying thresholds do not affect their metrics. For the best and simple models (**Fig 7E**; green bars and blue dots, respectively), lower thresholds increased the number of unnecessary isolation days, though notably the best model always had fewer days than the simple model. Lower thresholds also resulted in substantially fewer non-isolated infectious days, and both the simple and best models can outperform the five-day protocol (**Fig 7F**). Notably, lowering this threshold reduced the number of rebound individuals that are prematurely released from isolation. Although the simple model appeared to outperform the best model on the number of infectious days, this reduction actually resulted from the simple model failing to identify a second consecutive negative more often than the best model for all threshold values (e.g., 64.6% vs. 51.9% of individuals for a 10% threshold). This causes more individuals to default (by our assumption) to the ten-day procedure, thus also decreasing the number of non-isolated infectious days. Overall, the best model provides the most accurate and customizable approach–offering the potential to tune predictions to minimize non-isolated infectious days or to minimize unnecessary isolation days, depending on context and local priorities.

## Discussion

In this study, we developed a generalizable model to infer the results of one virological assay from another. By applying this framework to our compiled database of non-human primate experiments on SARS-CoV-2, we generated highly accurate predictions of sgRNA and culture results from standard PCR protocols. These analyses allowed us to answer foundational questions about whether totRNA and sgRNA assays are fundamentally interchangeable and what factors drive the complicated relationships between PCR and culture outcomes. Our best models identify key sources of biological and methodological variation (including exposure conditions, demographics, and assay protocols), across which predictions varied widely. We showed that because standard, single regression models (like our 'simple models') ignore this variation, they could incorrectly infer culture outcomes for samples with totRNA copy numbers spanning twelve orders of magnitude; our biologically-informed multiple regression models showed substantial gains in accuracy and precision. Our findings highlight the importance of accounting for the influence of cofactors on viral load and culture positivity–no single threshold value applies across study designs.

We addressed the unresolved debate about the relative merit of sgRNA to predict culture outcomes by conducting the first comprehensive analysis of a large dataset of controlled

exposures. We found no clear evidence that sgRNA outperforms totRNA, and instead we found that both infer culture outcomes with high accuracy when accounting for key biological covariates. Given these results and that we can reconstruct sgRNA trajectories from totRNA outcomes with high accuracy, underlying cofactors may explain previously observed differences in the relative predictive capacity of totRNA and sgRNA [10,14]. Future studies could prospectively measure all three assays (ideally with quantitative culture) to confirm and extend our findings, though notably our model achieved a remarkable 85% accuracy in predicting culture outcomes and our error analysis showed that many prediction errors may have arisen from upstream data issues (see below).

Our models characterize many biological patterns hypothesized (or known) based on previous experimental work on SARS-CoV-2, including the effects of exposure conditions on sgRNA and culture outcomes. In particular, we find that larger exposure doses increase the totRNA copy numbers associated with predicting culture positivity and detectable sgRNA. This suggests that the amplification of residual (inoculum-derived) genomic RNA may explain curious instances of sgRNA- or culture-negative samples with large totRNA copies, substantiating concerns in the animal challenge literature that inoculation procedures can directly influence viral detection and quantification [7]. Interestingly, when we included a lab effect, our best sgRNA model predicted that (for any given totRNA quantity) larger doses would increase or have no effect on sgRNA quantities. This pattern could arise from two dueling effects of the inoculation procedure, whereby larger doses may increase (at least initial) sgRNA production, but inoculum-derived and newly produced gRNA could mask this effect. Future experimental work could test this hypothesis by directly comparing a range of doses.

The amplification of residual inoculum may also explain differences predicted between inoculated and non-inoculated tissues, where exposed tissues tend to have larger totRNA quantities than non-exposed tissues for any given sgRNA value, particularly on the first day post infection. Inoculum effects on totRNA quantity appear to linger throughout infection, given that sgRNA predictions for exposed tissues on later days post infection fall between predictions for exposed tissues on the first day and non-exposed tissues on all days. Interestingly, the chance of positive culture (for a given totRNA value) is highest for exposed tissues sampled on the first day post infection, which is consistent with detection of lingering inoculum-derived virions. In contrast to sgRNA, culture predictions for exposed tissues on all later days post infection are highly similar to non-exposed tissues. These patterns are consistent with most inoculated virions having infected cells, dispersed to other tissues, or been cleared by the immune system within the first two days of infection, whereas the high stability of RNA (at least in human respiratory fluids monitored *ex vivo* [67]) could enable its prolonged detection.

Our work showed that the relationships between virological assays were also shaped by host demographic factors. Primate species affected all relationships we considered, where cynomolgus macaques were predicted to have the lowest sgRNA:totRNA ratio and the smallest chance of positive culture per totRNA quantity. African green monkeys and rhesus macaques have highly similar predictions for sgRNA:totRNA ratios and chances of positive culture. Curiously, African green monkeys also have the smallest chance of sgRNA detection per totRNA quantity, but only one study [8] reported totRNA and sgRNA outcomes for this species. Our models did not identify age-mediated effects on sgRNA outcomes but did predict that geriatric animals have the highest chances of positive culture per totRNA quantity. Sex did not influence either sgRNA or culture outcomes. While these results may reflect differing susceptibility, disease severity, or infection kinetics among non-human primate species and age classes, as has been previously suggested [26,28,43,58,59,68], sample sizes were limited for African green monkeys and geriatrics, so these patterns should be interpreted cautiously. Also, given the complexity of viral fitness, cellular processes, and immune responses, inference on the cause of

demographic-specific differences is difficult without mechanistic theory. Mathematical models of the cellular life cycle [69] may uncover processes that explain the stoichiometric differences we observed among RNA types and virions.

Assay protocols had clear impacts on model predictions. PCR target gene was a consistent factor in our best models, with effects aligned with known differences in RNA quantities. We find that totRNA protocols targeting the Spike (S) gene must amplify less totRNA than those targeting the Envelope (E) or Nucleocapsid (N) genes to predict the same chance of positive culture. This likely reflects that totRNA assays targeting S will amplify only sgS and no other sgRNA species (because it is the most upstream sgRNA), whereas the others amplify multiple sgRNA species and thus will have inherently higher per-sample totRNA copy numbers. Notably, this result does not imply that spike assays better predict infectivity. Different genes simply require different RNA quantities to expect the same chance of culture positivity, and so other considerations should motivate choice of target gene (e.g., selecting target sequences that are conserved across variants). Similar reasoning can explain observed differences in sgRNA outcomes, where sgRNA protocols amplifying the highly-expressed sgN have higher chances of detecting sgRNA (per totRNA quantity) and also larger sgRNA:totRNA ratios than protocols amplifying the less-expressed sgE and sg7 species. For viral culture, our model predicts VeroE6-TMPRSS2 cells have the highest chance of detecting infectious virus (per totRNA quantity), which is concordant with the importance of TMPRSS2 for SARS-CoV-2 cellular entry [62] and agrees with experiments showing VeroE6-TMPRSS2 cells are more permissive to infection than VeroE6 cells [21,61]. In accordance with our results, prior work has also shown that VeroE6 cells are more sensitive than Vero76 cells, which is likely related to increased TMPRSS2 expression in VeroE6 cells [70]. Our model also predicts that TCID50 assays are more likely to detect infectious SARS-CoV-2 than plaque assays, agreeing with standard assay conversions [71] and prior experimental work [63].

Although we developed this model to analyze SARS-CoV-2 in non-human primates, our results showed many similarities with patterns previously noted in humans. Multiple studies have found that, depending on the dataset, human-derived samples with around 5–9 log10 RNA copies had a 50% chance of being culture positive [6,19,28,72]. The prediction from our analogous model without cofactors falls within this range (7 log10 totRNA copies). Other work has found evidence of age-dependent increases in infectious virus shedding [73] or in culture probability on any day rescaled to the time since peak viral load [28]. Both of these findings are consistent with, although not directly comparable to, our result that geriatric NHPs have higher probabilities of culture positivity per totRNA quantity. Another study also discovered that the ratios of RNA to culturable virus differed substantially throughout infection [74]. We unfortunately did not have sufficient quantitative culture information to obtain a similar ratio, but their findings agree with our observation that (for any given totRNA quantity) sgRNA copy numbers and culture probability vary by day post infection. Finally, we observed no culture positive (non-invasive) samples from the respiratory tract more than seven days after an individual's first positive test, and so the public health guidelines of isolating for five or ten days [31] performed remarkably well on our dataset, despite being designed for an entirely different host species. Collectively, these concordances further underscore that non-human primates are an excellent model system for human SARS-CoV-2 infection.

By analyzing our culture predictions for individual trajectories, we identified potential causes of prediction errors. Many occurred during transition periods when viral replication slows or begins (i.e., when infectivity changes). During this crucial phase, our best culture model clearly outperformed the simple model by making fewer mistakes. In any case, during these periods, assay readouts will depend strongly on sample quality and assay sensitivity, so additional caution in interpreting culture outcomes is warranted. Beyond this, while we expect

some errors due to complex and non-stationary biological effects, many errors are also consistent with PCR or culture processing issues. Sample quality, preservation methods, and storage conditions can substantially impact the quantification of RNA copy numbers and the detection of infectious virus [75,76]. PCR issues resulting in the amplification of less RNA may explain curious culture-positive samples with low or no detectable RNA (generating false negative predictions), while culture insensitivity may explain some culture-negative samples with especially large RNA quantities (i.e., false positives). Alternatively, sample contamination or sample swapping could cause elevated RNA levels or spurious culture positivity, where the latter is particularly plausible for 'data blips' of a single culture positive surrounded by a series of culture negatives, although these could reflect brief, intermittent replication. In any case, if we assume our model predictions were correct for at least some of these suspect samples (or else if we exclude them from accuracy calculations entirely), our culture model's true accuracy would be higher than 85%.

With this study, we demonstrated the utility and feasibility of meta-analyses and Bayesian statistical techniques for virological studies, which will become increasingly important tools under new data sharing mandates [77]. Multiple factors enabled us to rigorously analyze our aggregate database: (i) PCR results were reported as RNA copy numbers, which are internally standardized (as opposed to unstandardized Ct values) [75], (ii) processing techniques and viral concentrations per reported sample volume are consistent within each study, (iii) many articles reported results for multiple cofactors, and (iv) we accounted for any additional between-study variation by including article-level hierarchical error rates when possible. To evaluate whether any of the observed patterns could be explained by unmodelled methodological differences among articles, we also ran our best models with an additional predictor for lab effects. Reassuringly, we found that all of our results were qualitatively unchanged between the models with and without lab effects (with one minor exception, discussed above), offering confidence in the robustness of our results. Under typical analytical approaches, our investigations would have required one study to generate the data for all protocols, samples, and demographics of interest, which would be time and resource prohibitive. Crucially, our approach did not require the generation of new data, which is especially important for non-human primate models where ethical principles [78,79] and constrained supply [80,81] demand principled data reuse whenever possible.

Although the concordances noted between prior work and our results offer confidence in our models' performance, our study has limitations. Multiple source articles did not report age class or sex, requiring our model fits to marginalize over all possibilities. Consequently, parameter estimates for age and sex may underestimate their effects. This underscores the importance of comprehensive reporting, especially for animal challenge experiments where using previously collected data would increase adherence to the 3R principles [78]. Also, few articles reported results for both sgRNA and culture, so some of our investigations relied on imputed sgRNA values. Prospective data on all three assays and more comprehensive data panels across cofactors would enable deeper exploration of the predictive capacity of totRNA and sgRNA for viral culture. Finally, while some cofactors were not selected for inclusion in our best models, we cannot exclude the possibility that their effects exist but were not evident or were masked by other predictors. Because covariate coverage relied on different studies in different labs, it remains possible that lab or study effects impacted our results even though we found no evidence of this when including lab-specific predictor variables. Some covariate effects may have also been absorbed into our article-level error or lab effect terms. Despite these limitations, our analysis (and similar analyses) can help prioritize resource allocation, so future experiments can more easily adopt the gold-standard approach of testing model-based findings in head-to-head comparisons under fixed conditions.

While the quantitative results of our models should not be used directly to predict culture results for any host-pathogen system besides non-human primates and SARS-CoV-2, the general framework could be adapted easily to generate similar predictions for other host species, other viruses, or other assays. For example, our model could be modified to robustly compare the relationships among antigen tests, PCR, and viral culture, which has recently garnered interest [14,15,82,83] and would benefit from the increased sample size and cofactor coverage possible with meta-analytical treatment. Notably, when applying the framework to other scenarios, careful model development is still necessary, especially given that different viruses and assays may have other defining characteristics that could affect their relationships, which should influence the choice of candidate cofactors.

We believe our framework also shows particular promise for future development to support clinical diagnostics. Beyond the fact that our model trained on NHP data recapitulated many patterns previously observed in humans, we also demonstrated its excellent performance on clinically relevant metrics. Relative to the five- or ten-day isolation protocols outlined by public health agencies [31], our best model substantially reduced unnecessary isolation time (relative to the ten-day rule), and it reduced the risk of releasing individuals while still infectious (relative to the 5-day rule). Our best model also clearly outperformed the simple model on both of these metrics, in addition to correctly classifying more sequential culture negative samples and with markedly higher confidence, all of which could be crucial improvements in public health settings. In fact, because sampling frequency decreased over the course of infection in our data, our results likely underestimate the potential improvements achievable in humans where sampling can be more frequent. To realize the clinical potential of this approach, however, the model framework must be trained on human data. This would involve some model modifications, including the consideration of other cofactors such as viral variant, prior infection, vaccination history, disease severity, and co-morbidities. Outside the very rare context of human challenge trials, the model will also need to function without knowledge of exposure dose, route, or exact timing (requiring the use of a proxy such as time since symptom onset or first positive test). If such a model performs well, then it would offer a straightforward, standardized pipeline to predict whether an individual is infectious based on SARS-CoV-2 PCR results, which is a clear need [9,17,19,21–24]. To further increase prediction accuracy, future work could also modify the framework to capitalize on individual-specific trajectories for patients undergoing regular screening (e.g., by incorporating a mechanistic modeling component [73]). Once the modeling pipeline is established, it could be readily tailored to any other pathogen with sufficient clinical data, either to improve management strategies of existing viruses or even to help characterize and contain an emerging one. With these tools, public health officials and clinicians would be better-equipped to weigh transmission risk with medical resource availability and economic burden to designate evidence-based (and pathogen-specific) hospital discharge criteria and public health guidelines.

By assembling and analyzing a large database that captures infection patterns described in the clinical and animal challenge literature, we demonstrated that highly accurate RNA-based culture predictions are possible with our statistical framework. By using non-human primate data, we were able to identify underlying effects of exposure conditions, which would be impossible for humans without experimental challenge trials (of which only one exists for SARS-CoV-2, to date [84]). Consequently, our model offers the first set of explicit quantitative guidelines on interpreting SARS-CoV-2 assay outcomes in light of exposure conditions, which has direct implications for analyzing non-human primate experiments and thus could affect human health by improving interpretations of crucial preclinical trials for human vaccines and therapeutics. We propose our method as a standardized framework to conduct assay comparisons, whether for individual virology experiments, clinical diagnostic settings, qualitative

literature syntheses, or quantitative meta-analyses. Such approaches for data aggregation and (meta-)analysis are vital and powerful tools for an era of increasing data-sharing, with untapped potential to develop translational applications and to guide further research into fundamental mechanisms.

## Supporting information

**S1 Methods. Additional methodological detail, including database compilation, prior justifications, performance analysis, model selection, and prediction generation.**
(PDF)

**S1 Fig. Screening and selection procedure for database compilation.** We created this figure by adapting the template flowchart provided in Moher et al. 2009 (34), which offers guidelines and resources for systematic reviews and meta-analyses. We incorporated all of their suggested steps for reporting the results of systematic literature searches, but all of the substantive content (e.g., numbers, exclusion reasons) is based entirely on our literature search. Additional detail on the screening procedure is provided in the S1 Methods.
(TIF)

**S2 Fig. Schematic diagram of generalizable hurdle model predicting assay Y from a more sensitive assay X.** Predictors are grey, model components are green, and predictions are red (positive) or blue (negative). If assay X falls below the limit of detection ($<$ LOD), assay Y is also predicted to fall below the limit of detection. (Note that this particular assumption may not hold for all assay relationships, and modeling adjustments may need to be made in these scenarios.) If assay X falls above the limit of detection ($>$ LOD), then the value of assay X is passed as a predictor to the logistic component of the hurdle model, which uses a set of additional covariates $A_i$ to predict whether assay Y falls above or below the LOD. If the posterior probability of assay Y falling above the limit of detection is less than some assigned threshold C ($P(Y > LOD) < C$), then the model predicts assay Y falls below the LOD. Otherwise, the model predicts assay Y falls above the LOD. Note that the probability cut-off value C should be selected to balance false positive and false negative rates as appropriate to investigator aims. In this study, we used a standard value of C = 0.5. For samples predicted to fall above the LOD, the linear model component will generate a predicted value of assay Y ($Y_{predict}$) based on another set of covariates ($B_j$). If $Y_{predict}$ is larger than the reported LOD for assay Y, the model will return the predicted value. Created with BioRender.com.
(TIF)

**S3 Fig. Individual viral load trajectories in the upper respiratory tract, including sgRNA predictions generated by the best sgRNA model.** Each panel corresponds with one individual and one non-invasive sample type, indicated in the top right of each panel. Only individuals with both total RNA and sgRNA results for at least two days post infection are plotted. Some individuals were sampled from multiple locations in the upper respiratory tract, in which case they are plotted as neighboring panels. Each line and the accompanying points track the individual's total RNA (dark blue, circle), observed sgRNA (light blue, diamond), and median predicted sgRNA (green, triangle) trajectories. For some individuals (e.g., KS_2021C), multiple RT-qPCR assays targeting different genes were run on the same sample, which are plotted as distinct panels. All samples observed or predicted to fall below the limit of detection are plotted below 0 at set values for visual clarity (totRNA: -0.5, observed sgRNA: -0.75, predicted sgRNA: -1). When available, the limits of detection (LOD) or quantification (LOQ) for PCR assays are plotted as dotted lines in the assay-specific color. When both the LOD and LOQ were available, only the LOD is plotted. In instances where the total RNA and sgRNA assay LOD are equal,

only the sgRNA line is visible. No instances exist in this dataset where the LOD or LOQ is only available for one RNA type.
(TIF)

**S4 Fig. Individual viral load trajectories in the lower respiratory tract, including sgRNA predictions generated by the best sgRNA model.** Each panel corresponds with one individual and one non-invasive sample type, indicated in the top right of each panel ('BAL': bronchoal-veolar lavage). Only individuals with both total RNA and sgRNA results for at least two days post infection are plotted. Each line and the accompanying points track the individual's total RNA (dark red, circle), observed sgRNA (orange, diamond), and median predicted sgRNA (yellow, triangle) trajectories. For some individuals (e.g., KS_2021C), multiple RT-qPCR assays targeting different genes were run on the same sample, which are plotted as distinct panels. All samples observed or predicted to fall below the limit of detection are plotted below 0 at set values for visual clarity (totRNA: -0.5, observed sgRNA: -0.75, predicted sgRNA: -1). When available, the limits of detection (LOD) or quantification (LOQ) for PCR assays are plotted as dotted lines in the assay-specific color. When both the LOD and LOQ were available, only the LOD is plotted. In instances where the total RNA and sgRNA assay LOD are equal, only the sgRNA line is visible. No instances exist in this dataset where the LOD or LOQ is only available for one RNA type.
(TIF)

**S5 Fig. Individual viral load trajectories in the gastrointestinal and other systems, including sgRNA predictions generated by the best sgRNA model.** Each panel corresponds with one individual and one non-invasive sample type, indicated in the top right of each panel. Only individuals with both total RNA and sgRNA results for at least two days post infection are plotted. Each line and the accompanying points track the individual's total RNA (dark purple, circle), observed sgRNA (dark pink, diamond), and median predicted sgRNA (light pink, triangle) trajectories. All samples observed or predicted to fall below the limit of detection are plotted below 0 at set values for visual clarity (totRNA: -0.5, observed sgRNA: -0.75, predicted sgRNA: -1). When available, the limits of detection (LOD) or quantification (LOQ) for PCR assays are plotted as dotted lines in the assay-specific color. When both the LOD and LOQ were available, only the LOD is plotted. In instances where the total RNA and sgRNA assay LOD are equal, only the sgRNA line is visible. No instances exist in this dataset where the LOD or LOQ is only available for one RNA type.
(TIF)

**S6 Fig. Individual viral loads for invasive samples, including sgRNA predictions generated by the best sgRNA model.** Each panel corresponds with one individual, indicated with text in the panel (day post infection: individual). Each point presents the total RNA (circle), observed sgRNA (diamond), and predicted sgRNA (triangle) values. All samples observed or predicted to fall below the limit of detection are plotted below 0 at set values for visual clarity (totRNA: -0.5, observed sgRNA: -0.75, predicted sgRNA: -1). When available, the limits of detection (LOD) or quantification (LOQ) for PCR assays are plotted as dotted lines in the assay-specific color. When both the LOD and LOQ were available, only the LOD is plotted. In instances where the total RNA and sgRNA assay LOD are equal, only the sgRNA line is visible. No instances exist in this dataset where the LOD or LOQ is only available for one RNA type.
(TIF)

**S7 Fig. Individual culture trajectories in the upper respiratory tract.** Each panel corresponds with one individual and one non-invasive sample type, indicated in the top right of each panel. Only individuals with culture results for at least two days post infection are plotted.

Culture data are plotted as squares above the yellow line at 10 log10 copies. Yellow squares are culture positive samples, while grey squares are culture negative. Squares outlined in black are correct predictions, squares with no outline are incorrect predictions. We did not generate predictions for the culture samples outlined in blue, as they do not have available totRNA results. We also plot observed total RNA values (circle) and observed sgRNA values (diamond), otherwise we plot predicted median sgRNA values generated by our best sgRNA model (triangle). Some individuals were sampled from multiple locations in the upper respiratory tract, in which case they are plotted as neighboring panels. All samples observed or predicted to fall below the limit of detection are plotted below 0 at set values for visual clarity (totRNA: 0, sgRNA: -1). When available, the limits of detection (LOD) or quantification (LOQ) for PCR assays are plotted as dotted lines in the assay-specific color. When both the LOD and LOQ were available, only the LOD is plotted. In instances where the total RNA and sgRNA assay LOD are equal, only the sgRNA line is visible. No instances exist in this dataset where the LOD or LOQ is only available for one RNA type. Individuals from one study cannot be included in this figure due to a data sharing agreement.
(TIF)

**S8 Fig. Individual culture trajectories in the lower respiratory tract.** Each panel corresponds with one individual and one non-invasive sample type, indicated in the top right of each panel. Only individuals with culture results for at least two days post infection are plotted. Culture data are plotted as squares above the yellow line at 10 log10 copies. Yellow squares are culture positive samples, while grey squares are culture negative. Squares outlined in black are correct predictions, squares with no outline are incorrect predictions. We did not generate predictions for the culture samples outlined in blue, as they do not have available totRNA results. We also plot observed total RNA values (circle) and observed sgRNA values (diamond) when available, otherwise we plot predicted median sgRNA values generated by our best sgRNA model (triangle). Some individuals were sampled from multiple locations in the lower respiratory tract, in which case they are plotted as neighboring panels. All samples observed or predicted to fall below the limit of detection are plotted below 0 at set values for visual clarity (totRNA: 0, sgRNA: -1). When available, the limits of detection (LOD) or quantification (LOQ) for PCR assays are plotted as dotted lines in the assay-specific color. When both the LOD and LOQ were available, only the LOD is plotted. In instances where the total RNA and sgRNA assay LOD are equal, only the sgRNA line is visible. No instances exist in this dataset where the LOD or LOQ is only available for one RNA type.
(TIF)

**S9 Fig. Individual culture trajectories in the gastrointestinal and other systems.** Each panel corresponds with one individual and one non-invasive sample type, indicated in the top right of each panel. Only individuals with culture results for at least two days post infection are plotted. Culture data are plotted as squares above the yellow line at 10 log10 copies. Yellow squares are culture positive samples, while grey squares are culture negative. Squares outlined in black are correct predictions, squares with no outline are incorrect predictions. We also plot observed total RNA values (circle) and observed sgRNA values (diamond) when available, otherwise we plot predicted median sgRNA values generated by our best sgRNA model (triangle). Some individuals were sampled from multiple locations, in which case they are plotted as neighboring panels. All samples observed or predicted to fall below the limit of detection are plotted below 0 at set values for visual clarity (totRNA: 0, sgRNA: -1). When available, the limits of detection (LOD) or quantification (LOQ) for PCR assays are plotted as dotted lines in the assay-specific color. When both the LOD and LOQ were available, only the LOD is plotted. In instances where the total RNA and sgRNA assay LOD are equal, only the sgRNA line is

visible. No instances exist in this dataset where the LOD or LOQ is only available for one RNA type. Individuals from one study cannot be included in this figure due to a data sharing agreement.
(TIF)

**S10 Fig. Individual culture data for invasive samples.** Each panel corresponds with one individual, indicated with text in the panel (day post infection: individual). Culture data are plotted as squares above the yellow line at 10 log10 copies. Yellow squares are culture positive samples, while grey squares are culture negative. Squares outlined in black are correct predictions, squares with no outline are incorrect predictions. We did not generate predictions for the culture samples outlined in blue, as they do not have available totRNA results. We also plot the observed total RNA (circle) and observed sgRNA (diamond) values when available, otherwise we plot predicted median sgRNA values generated by our best sgRNA model (triangle). Color corresponds to the organ system from which the tissue was obtained (URT, upper respiratory tract; LRT, lower respiratory tract; GI & Other, gastrointestinal and other systems). All samples observed or predicted to fall below the limit of detection are plotted below 0 at set values for visual clarity (totRNA: 0, sgRNA: -1). When available, the limits of detection (LOD) or quantification (LOQ) for PCR assays are plotted as dotted lines in the assay-specific color. When both the LOD and LOQ were available, only the LOD is plotted. In instances where the total RNA and sgRNA assay LOD are equal, only the sgRNA line is visible. No instances exist in this dataset where the LOD or LOQ is only available for one RNA type.
(TIF)

**S11 Fig. Statistics relating PCR and culture results.** (A) Difference between total RNA and sgRNA copy numbers when both are detectable, stratified by target gene predictor with the following acronyms: "T↑SG↑": totRNA-high/sgRNA-high; "T↓SG↑": totRNA-low/sgRNA-high; "T↑SG↓": totRNA-high/sgRNA-low; "T↓SG↓": totRNA-high/sgRNA-low. No totRNA-high/sgRNA-high data was available for this investigation. (B) Total RNA copy numbers for all sgRNA negative samples, stratified by target gene as in (A). (C) Pearson correlation coefficients between total RNA and sgRNA copy numbers when both are detectable, for all individual-sample trajectories with at least three sampling days where both were positive. (D) Comparison of the timing of the first negative results from total RNA and sgRNA assays for each available individual-sample trajectory (dpi: day post infection). (E) Total RNA copy numbers (when detectable) for all culture positive samples, stratified by culture assay type. (F) Total RNA copy numbers (when detectable) for all culture negative samples, stratified by culture assay type as in (E). (G) Comparison of the timing of the first negative results from total RNA and culture assays for each available individual-sample trajectory. (H) Comparison of the timing of the first positive results from total RNA and culture assays for each individual-sample trajectory. For panels (A), (B), (C), (E), and (F), the purple dashed line indicates the median for the full distribution (i.e., not stratified by assay or target gene). For panels (D), (G), and (H), the size of each circle indicates the number of individuals with the indicated observation. Individuals in the 'None' column were never negative (D, G) or positive (H) for total RNA. Individuals that were never sgRNA negative (D), culture negative (G), or culture positive (H) are not plotted.
(TIF)

**S12 Fig. Results from the best sgRNA model with an additional predictor for lab group.** (A) The predicted chances of sgRNA detection for three key totRNA quantities (3 log10, blue; 5 log10, salmon; 7 log10, red), across the eight available lab groups and for the standard cofactor set. The article(s) included in each group are provided in S8 Table. Each point is one out of

200 samples generated for each lab group, with transparency to show the density of points. (B) As in Fig 3B, with additional predictions from the model including a lab effect ('Lab', grey). (C and D) As in Fig 3C and 3D, except showing the results from the model including a lab effect. (E) The predicted quantities of sgRNA for a sample with 5 log10 totRNA copies, across the eight available lab groups and for the standard cofactor set. (F) As in Fig 3F, with additional predictions from the model including a lab effect ('Lab', grey). (G and H) As in Fig 3G and 3H, except showing the results from the model including a lab effect. In panels C, D, G and H, the predictions are not specific to a particular lab group (i.e., we set the lab effect term to zero to extract general patterns across all labs).
(TIF)

**S13 Fig. Sensitivity analyses comparing informative (blue) and non-informative (red) priors.** (A) Each line presents an expected model fit generated by sampling the indicated prior distributions. Informative priors are outlined in the **Methods** and **S1 Methods**. All parameters were given a $N(0,1)$ prior for all non-informative investigations. Informative priors much better represent *a priori* understanding of the relationships between total RNA copy numbers and both sgRNA and culture outcomes. (B) Each panel compares the final parameter estimates obtained for the corresponding best model using the different prior types (red: non-informative; blue: informative), where each row is a distinct parameter. Acronyms are as described in **Figs 3** and **5**. Note that in many instances parameters estimates are almost perfectly overlapping, so only the non-informative (red) priors are visible.
(TIF)

**S14 Fig. Error analysis for the best sgRNA model.** (A) Individual-specific sgRNA trajectories, where each row presents one individual. These are stratified by whether the model misclassifies any samples for that individual ("Some errors") or whether the model makes no misclassifications ("No errors"). In both (A) and (B), yellow circles indicate positive samples and grey indicates negative samples. Circles with a black outline correspond with correctly classified samples, while no outline indicates incorrectly classified samples. (B) Scatterplot of all samples with sgRNA results, stratified by the elements of a confusion matrix and colored as in (A). The x-axis tracks the day post infection and the y-axis plots log10 total RNA copy numbers. Samples in the grey shaded region along the bottom present all samples where total RNA was undetectable. (C) Histograms of all samples grouped by the elements of a confusion matrix, where log10 total RNA copy numbers per sample is plotted on the y-axis. Bins located in the grey shaded region along the bottom (labelled "<LOD") include all totRNA-negative samples.
(TIF)

**S15 Fig. Additional performance comparisons between the simple and best culture models.** (A) Distribution of the differences between the predicted probabilities of both models for all totRNA-positive samples, stratified by whether the sample was culture positive (yellow) or negative (grey). Samples on the right side of the dashed blue line were predicted with higher confidence by the best model, while those on the left side were predicted with higher confidence by the simple model. (B) Distribution of median model-predicted chances of positive culture for intermediate totRNA-positive samples (6–8 log10 copies), stratified by model type and observed outcomes. Samples right of the dashed vertical line are correct predictions. The colored text gives the percent of samples that are correctly classified by each model. (C) As in panel A, except only for intermediate totRNA-positive samples (6–8 log10 copies).
(TIF)

**S16 Fig. Results from the best culture model with an additional predictor for lab group.**
(A) The predicted chances of culture positivity for three key totRNA quantities (3 log10, blue; 7 log10, salmon; 11 log10, red), across the ten available lab groups and for the standard cofactor set. The article(s) included in each group are listed in S8 Table. Each point is one out of 200 samples generated for each lab group, with transparency to show the density of points. (B) As in Fig 5B, with additional predictions from the model including a lab effect ('Lab', grey). (C and D) As in Fig 5C and 5D, except showing the results from the model including a lab effect. In panels C and D, the predictions are not specific to a particular lab group (i.e., we set the lab effect term to zero to extract general patterns across all labs).
(TIF)

**S17 Fig. Viral load and culture trajectories for individuals with data blip (A) or prediction blip (B) error types.** Panel-specific errors are indicated with red outlines. All other samples with prediction errors have no outline. Correct predictions are outlined in black. Yellow squares indicate known culture positive samples, while grey squares indicate known culture negative samples. Text in the upper right corner of each panel indicates the ID name and sample type of the individual from whom the data was derived. All totRNA-negative samples are plotted below the grey dashed line at zero. Note that individual NN_#5412 has an additional (true negative) sample available on a later day post infection, which is not shown for visual clarity. Six trajectories from one study cannot be included in this figure due to a data sharing agreement.
(TIF)

**S18 Fig. Isolation end times predicted by the simple (A) and best (B) culture models.** Each row is a unique individual, and each panel displays all individuals included in the isolation analyses. The results of all samples after every individual's first positive test (PCR or culture) are displayed, where culture positive samples are yellow and negative samples are grey. Each individual's last culture positive and their subsequent culture negative times are plotted with more intensity for better visualization. For each individual, their isolation end time is shown with colored, filled diamonds (i.e., the time of their second consecutive predicted culture negative test). When isolation end time could not be determined by the model (i.e., the model did not predict a second consecutive negative), we conservatively set that individual's end time to day 10. Each individual's first predicted negative is shown by an empty diamond, and the true (observed) time of their second consecutive negative is shown with a small red point. With yellow lines, we show the time range that we consider each individual to be infectious, based on the data, which ranges from their first total RNA positive day up to the midpoint between their first culture negative test after their last observed culture positive test. For individuals with no observed negative after their last positive, we conservatively assumed their next observed negative to be day 10. With dashed red lines, we also indicate which individuals show evidence of a rebound infection (i.e., the individuals with at least one culture negative occurring between two culture positives). Finally, we use colored vertical lines to display the days on which the five- and ten-day protocols would release individuals from isolation.
(TIF)

**S19 Fig. Days between consecutive tests relative to the number of days since the first positive test.** The size of the point shows the number of samples at the given coordinate. The marginal histograms show the distribution of points along each individual axis.
(TIF)

**S1 Table. Summary of articles included in the dataset.** Multiple rows for an individual article are included when the study involved multiple species and/or multiple exposure doses. In all

columns, U indicates the detail is unknown. Sample sizes (N) are presented in the following format: number of available datapoints (number of individuals). Species abbreviations are as follows: RM, rhesus macaque; CM, cynomolgus macaque; AGM, African green monkey. Age class presents the standardized assignments according to our protocol (**S1 Methods**), and the abbreviations are: J, juvenile; A, adult; G, geriatric. Individuals inoculated via multiple routes are indicated by exposure routes joined by commas, where the abbreviations are: AE, aerosol; IT, intratracheal; IN, intranasal; IG, intragastric; OC, ocular; OR, oral. Exposure dose is presented as log10 plaque forming units, and an adjoining * indicates the dose was originally reported as TCID50, so those values were converted using the standard method described in the **S1 Methods.** NI indicates non-invasive sample types (i.e., swabs, biofluids, BAL), while I indicates invasive tissue samples obtained at necropsy. Sample location distinguishes between the following systems: URT, upper respiratory tract; LRT, lower respiratory tract; GI, gastrointestinal tract; and Other, all other locations. Sample time presents the days post infection with available samples according to our DPI predictor, where 1: 1 dpi, inoculated tissues, 2: 2+ dpi, inoculated tissues, 3: any dpi, non-inoculated tissue (further categorization information is in **S9 Table**). PCR target genes are stratified by total RNA (totRNA) and sgRNA. The level of the target gene predictor for the sgRNA model follows the sgRNA gene in parentheses: (1) totRNA-high/sgRNA-high, (2) totRNA-low/sgRNA-high, (3) totRNA-high/sgRNA-low, and (4) totRNA-low/sgRNA-low. The cell lines used for culture are indicated when available, with SS2 as an abbreviation for TMPRSS2. An adjoining † indicates the use of a TCID50 assay, while no symbol indicates a plaque assay.
(DOCX)

**S2 Table. Extended sgRNA logistic model performance comparisons.** Models are ordered by increasing number of predictors, with the simplest (l1), best (l4.2), and full (l8.1) models noted in bold. We report expected log pointwise predictive density (ELPD) generated by 10-fold cross validation (cross-validation columns), where larger ELPD indicates better performance. ELPD difference indicates the difference between ELPDs of the given model and the model with the largest ELPD (in this case model l6.1, though this is not our 'best model'). The PSIS-LOO approximation columns present statistics generated by running Pareto-Smoothed Importance Sampling approximate leave-one-out cross validation, including ELPD and ELPD difference as above. The prediction columns indicate the percent of samples (stratified by training and test sets) for which posterior predictions generated by 10-fold cross validation correctly classified them as below or above the limit of detection (i.e., where the per-sample posterior predictive distributions exhibited at least a probability of 0.5 for the true, observed classification). MCC is the Matthews correlation coefficient. Note that all models included total RNA as a predictor, even though it is not specified in the predictor column. Standard error (SE) is shown in parentheses following all relevant statistics.
(DOCX)

**S3 Table. Extended sgRNA linear model performance comparisons.** Models are ordered by increasing number of predictors, with the simplest (f1), best (f5.1), and full (f8.1) models noted in bold. We report expected log pointwise predictive density (ELPD) generated by 10-fold cross validation (cross-validation columns), where larger ELPD indicates better performance. The best logistic model was run in tandem with all tested linear components, so the ELPD reported here reflects the sum of the ELPD for the best logistic and the considered linear components. ELPD difference indicates the difference between ELPDs of the given model and the model with the largest ELPD (in this case model l5.1, the 'best model'). The PSIS-LOO approximation columns present statistics generated by running Pareto-Smoothed Importance Sampling approximate leave-one-out cross validation, including ELPD and ELPD difference.

Standard error (SE) is shown in parentheses following all relevant statistics. We also used multiple metrics to assess model predictions, which are all stratified by performance on training versus test data sets and were generated by 10-fold cross validation. MAE is the median difference between the observed value and the posterior predictive median (i.e., median absolute error around the median) for all samples with sgRNA above the LOD, and this metric was also scaled by one standard deviation (Scaled). '% within 50% PI' and '% within 95% PI' columns indicate the percent of sgRNA positive samples where the true, observed value fell within the sample-specific 50% and 95% prediction intervals, respectively. Note that all models included total RNA as a predictor, even though it is not specified in the predictor column.
(DOCX)

**S4 Table. Extended culture model performance comparisons with totRNA as the primary predictor.** Models are ordered by increasing number of predictors, with the simplest (c1), best (c8.1), and full (c10.1) models noted in bold. We report expected log pointwise predictive density (ELPD) generated by 10-fold cross validation (cross-validation columns), where larger ELPD indicates better performance. ELPD difference indicates the difference between ELPDs of the given model and the model with the largest ELPD (in this case model l9.2, though this is not our 'best model'). The PSIS-LOO approximation columns present statistics generated by running Pareto-Smoothed Importance Sampling approximate leave-one-out cross validation, including ELPD and ELPD difference. The prediction column indicates the percent of samples (stratified by training and test sets) for which posterior predictions generated by 10-fold cross validation correctly classified them as below or above the limit of detection (i.e., where the per-sample posterior predictive distributions exhibited at least a probability of 0.5 for the true, observed classification). MCC is the Matthews correlation coefficient. Standard error (SE) is shown in parentheses following all relevant statistics.
(DOCX)

**S5 Table. Extended culture model performance comparisons with sgRNA as the primary predictor.** Models are ordered by increasing number of predictors, with the simplest (c1) and best/full (c10.1) models noted in bold. We report expected log pointwise predictive density (ELPD) generated by 10-fold cross validation (cross-validation columns), where larger ELPD indicates better performance. ELPD difference indicates the difference between ELPDs of the given model and the model with the largest ELPD (in this case model c10.1, our 'best model'). The PSIS-LOO approximation columns present statistics generated by running Pareto-Smoothed Importance Sampling approximate leave-one-out cross validation, including ELPD and ELPD difference. The prediction column indicates the percent of samples (stratified by training and test sets) for which posterior predictions generated by 10-fold cross validation correctly classified them as below or above the limit of detection (i.e., where the per-sample posterior predictive distributions exhibited at least a probability of 0.5 for the true, observed classification). MCC is the Matthews correlation coefficient. Standard error (SE) is shown in parentheses following all relevant statistics.
(DOCX)

**S6 Table. 90% prediction intervals for the best sgRNA model.** These intervals correspond with the predictions in Fig 3C and 3H.
(DOCX)

**S7 Table. Parameter estimates for the best models.** These were generated for the models without a lab effect.
(DOCX)

**S8 Table. Articles grouped into labs based on where the primate study was conducted.** The group number used to display lab effects in S12A and S12E, and S16A Figs are provided in the first column. The number for culture analyses (C) precedes the one for the sgRNA analyses (SG).
(DOCX)

**S9 Table. Performance comparison of culture models using totRNA, sgRNA, or both as the primary predictor(s).** Statistics are stratified by predictor(s) and the dataset used for fitting, including the full dataset (based on sgRNA predictions; 'all data') and the subset containing only samples with known sgRNA and totRNA results ('data subset'). Prediction accuracy reflects aggregate performance on test data across the full 10 train-test folds, stratified by all available samples (Overall), only known positive samples, and only known negative samples. MCC corresponds to the Matthews correlation coefficient. Note that we do not report ELPD because these models were fit with different quantities of data and so ELPD is not comparable. * includes imputed data. † includes data with observed sgRNA outcomes but no observed totRNA outcomes.
(DOCX)

**S10 Table. 90% prediction intervals for the best culture model.** These intervals correspond with the predictions in Fig 5C.
(DOCX)

**S11 Table. Categorization of inoculated versus non-inoculated sample locations per exposure route.** For every inoculation route, only the tissues with data available for that route are displayed. Because fluid is administered in the trachea for intratracheal (IT) inoculations, which is connected directly to the bronchioles, we include bronchus as an exposure tissue for IT inoculations. We also consider BAL an inoculated tissue for IT exposures since this procedure collects fluid from similar areas where the inoculum is administered. Exposure route abbreviations are: AE, aerosol; IT, intratracheal; IN, intranasal; IG, intragastric; OC, ocular; OR, oral.
(DOCX)

## Acknowledgments

We thank the authors of all articles included in this study, many of whom kindly corresponded with us to provide clarifications and raw data. We also thank the members of the Lloyd-Smith lab for valuable discussions of this work.

## Author Contributions

**Conceptualization:** Celine E. Snedden, James O. Lloyd-Smith.

**Formal analysis:** Celine E. Snedden.

**Funding acquisition:** Celine E. Snedden, James O. Lloyd-Smith.

**Methodology:** Celine E. Snedden, James O. Lloyd-Smith.

**Visualization:** Celine E. Snedden.

**Writing – original draft:** Celine E. Snedden.

**Writing – review & editing:** Celine E. Snedden, James O. Lloyd-Smith.

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
