## [Decision Letter · Decision Letter 0]

14 Dec 2023

Dear Dr. Lloyd-Smith,

Thank you very much for submitting your manuscript "PCR data accurately predict infectious virus: a meta-analysis of SARS-CoV-2 in non-human primates" for consideration at PLOS Pathogens. As with all papers reviewed by the journal, your manuscript was reviewed by members of the editorial board and by several independent reviewers. In light of the reviews (below this email), we would like to invite the resubmission of a significantly-revised version that takes into account the reviewers' comments.

Although the authors performed a detailed meta-analysis to predict infectious virus from PCR data, the reviewers (nr. 1 and nr. 2) and the editor found the lack of a more clear clinical application a major weakness of the study. The authors themselves mention in the discussion that "The framework also shows particular potential for clinical diagnostics, where it offers a straightforward, standardized pipelines to predict whether an individual is infectious based on PCR results, which is a clear need. .... This would initially require careful calibration on high-quality data and some model modifications, including adjusting for infection timing by accounting for days post symptom onset instead of days post infection ". The revision of the manuscript should mainly address this issue. In case, the proposed revision is not possible we only can provide the option of a transfer to PLOS Computational Biology. Please contact the staff and the editors if you would decide to do so.

We cannot make any decision about publication until we have seen the revised manuscript and your response to the reviewers' comments. Your revised manuscript is also likely to be sent to reviewers for further evaluation.

Sincerely,

Bart L. Haagmans

Academic Editor

PLOS Pathogens

Alexander Gorbalenya

Section Editor

PLOS Pathogens

Kasturi Haldar

Editor-in-Chief

PLOS Pathogens

orcid.org/0000-0001-5065-158X

Michael Malim

Editor-in-Chief

PLOS Pathogens

orcid.org/0000-0002-7699-2064

Although the authors perform a detailed meta-analysis to predict infectious virus from PCR data, the reviewers (nr1 and nr 2) and the editor found the focus on non human primate samples and lack of a clinical application a major weakness of the study. I am afraid that in the absence of evidence that (as stated by the authors in the discussion, line 600) " The framework also shows particular potential for clinical diagnostics..... ", I have to reject this version of the manuscript for publication in PLOS Pathogens.

Reviewer's Responses to Questions

**Part I - Summary**

Reviewer #1: The study used various experimental data in NHP models to predict infectious viruses.

Reviewer #2: In this manuscript, Snedden and Lloyd-Smith perform a meta-analysis of 24 studies and develop Bayesian statistical models to evaluate the utility of molecular markers as predictors of cell culture infectivity of SARS-CoV-2 from 3 species of experimentally infected non-human primates. The gold standard for infectivity of SARS-CoV-2 is demonstration of viral viability in cell culture. Molecular assays to detect infectivity would obviate the need for time consuming and expensive cell culture techniques that are not practical in a clinical setting. The authors are specifically interested in the utility of RT-qPCR assays of total SARS-CoV-2 RNA which represent the vast majority of RT-qPCRs tests used in a clinical setting. The authors also evaluate the utility of RT-qPCR assays of subgenomic RNA because early studies suggested that it may more closely reflect duration of infectivity. Subgenomic RNA assays, have not been used in clinical diagnostics despite multiple papers on the subject. The authors indicate their model can be adapted to support medical decisions and public health guidelines and generate predictions on infectiousness.

The authors results show that their models can predict subgenomic RNA results from total RNA and predict culture positivity from these two RT-qPCR assays with a high degree of accuracy. They explore multiple cofactors including host characteristics, and assays in their model and highlight the importance of these cofactors in culture prediction.

The strength of this study that the authors use a large number of data points under controlled conditions. The study is carefully conducted, and the data is fully explored, with extensive analysis, including in-depth exploration of reasons for unexpected patterns seen in the data. It does seem that this type of analysis would be a powerful tool especially if applied to the accumulating data from clinical samples and especially so in populations where duration of infectivity is less clear such as in immunosuppressed hosts or during infection with different viral variants. The manuscript is well written.

The study has some limitations which are mostly well outlined in the discussion. Because the authors indicate that their model can be used to support medical decisions and public health guidelines, they should address the difficulty in using rt-qPCR assays in clinical decision making. There is likely a copy number of sgRNA and TotRNA where patients are no longer infectious.

However, in clinical practice RT-qPCR assays have not generally been used to determine infectivity for many reasons. One reason is that it is very difficult to compare rt-qPCR results (either copy number or CT value) across labs without a universal calibrator for viral load. It also remains to be seen whether predictions models using Rt-qPCR data can improve the ability to predict infectivity allowing patients to be removed from in-hospital isolation in less than the current 10 days recommended for immunocompetent hosts. If possible, it may strengthen the manuscript to specifically indicate how the best model improves predictions on duration of infectivity when compared to the simple model.

Reviewer #3: Snedden and Lloyd-Smith investigated the relationship between total RNA, sub-genomic RNA and infectious culture positivity by compiling experimental data from 24 studies of SARS-CoV-2 infection in non-human primates. They developed Bayesian statistical models to quantify their relationships and evaluated how co-factors, such as exposure conditions, host traits, and assay protocols, influence model predictions. The main conclusion of the work is that total RNA as measured by PCR can predict culture positivity with a high accuracy and they did not find evidence that the sgRNA level is a superior predictor of culture positivity.

Overall, the work addresses an important question during the COVID-19 pandemic, i.e. how various measures of SARS-CoV-2 load could be used to estimate infectiousness of individuals. The model and results here will be very useful to understand the various assays and how their results could be used to make clinical or public health decisions in the pandemic. The work is well conducted, and well-written.

**Part II – Major Issues: Key Experiments Required for Acceptance**

Reviewer #1: The reviewer has the concerning concerns:

1. How does it relevant to human infection?

2. Considering most of the human individuals have prior immune responses, either by infection or vaccination, what is the potential use of their findings in human infections.

3. The authors used a vast number of NHP studies in the analyses. But for major outbreaks (or pandemics), it would be much easier to have clinical samples from infected cased to demonstrate "infectivity". How can their findings help to develop a better strategy to control/contain a disease in future?

4. Consider different respiratory virus infections are different clinical profile, the reviewer is not sure whether the authors intend to restrict the interpretations to COVID-19 or beyond. Please clarify.

Reviewer #2: #1:

Line 97

“Our results have many applications including: 1. to improve PCR-based clinical decisions with more accurate predictions of infectious status”

Unclear if study does this in a clinically significant way. For instance, would small gains in prediction accuracy percent seen in figure 2 (81% to ~85% for culture logistic) make a meaningful clinical difference (ie days to ending isolation). Does use of additional predictors decrease the error in predictions of days an animal is infectious such that it would have clear effect on duration of isolation. The goal of this paper seems to be an attempt to improve on current best practices by using a model to determine infectivity. If they can demonstrate this in a clinically meaningful way, it would strengthen the impact of this study.

#2:

I have concerns about the comparability of data across studies because even “absolute copy number” is likely relative to the particular standard used in each study. Interestingly, most studies used invitro transcribed RNA for sgE standards however, some did not. (**see below for method used in studies). It is not clear if the same invitro transcribed and quantitated RNA was used across studies. If the authors could explore the data to evaluate whether lab or study influences variation seen, this may alleviate concerns about comparing quantities across studies.

Related to above concern:

Lines 302-303: Referring to Figure 3E. Is it possible that “considerable unexplained variation” noted is an artifact of comparisons across studies where copy numbers are determined using different standards. In figure 3E for the linear comparison (red data points) it almost appears as there are 2 clusters of data with an upper cluster of data points and a parallel lower cluster of points. Do these apparent clusters correlate with different studies or labs?

Line 571-572 “multiple factors enable us to rigorously analyze our aggregate database: (i) …RNA copy numbers, which are internally standardized as opposed to unstandardized CT values (68).”

Caution is warranted in interpreting copy number reported as absolute and comparable. It is unlikely that a universal RNA standard was used across studies. Differences in invitro transcription, RNA extraction, and efficiency of qPCR reactions (to name a few variable) across laboratories could account for differences seen. The authors allude to this issue in the discussion Line 591-593: “…it remains possible that lab or study effects impacted our results…”

**Methods of generating standards

Baum: In vitro transcribed RNA for sgE standard

Chandrashekar: In vitro transcribed RNA for sgE standard

Corbert: In vitro transcribed RNA for sgE standard

Patel: In vitro transcribed RNA for sgE Standard

Salguero: In vitro transcribed of RNA for SgE standard

Li and Edwards: In vitro transcribed E and N

Dagotto: In vitro transcribed sgE for standard

Gabitzsch

“A standard curve comprised of synthetic RNA containing the corresponding target sequence from SARS-CoV-2 isolate WA1 sequence” Bio-synthesis Inc.

This synthetic RNA should not contain sgRNA but only gRNA and should not amplify with sgRNA primers. I suspect they used the gRNA standard to quantitate sgRNA but they use N primers for gRNA

Sparanza: “In each run, standard dilutions of counted RNA standards were run in parallel to calculate copy numbers in the samples”. Unclear if this was invitro transcribed or copy number determined using an alternative such as digital PCR.

Reviewer #3: None.

**Part III – Minor Issues: Editorial and Data Presentation Modifications**

Reviewer #1: See below

Reviewer #2: Line 30: Not sure clinical significance of infectivity of non-inoculated tissues. Other tissues (brain, spleen, intestines) lively have much less influence, if any, on infectivity compared to respiratory tract where virus is shed. Consider removing this example from author summary as non-inoculated samples are not sampled in clinical setting.

Line 32/33. Unclear what “quantitative predictions of infectiousness” means. Does this refer to viral load at which animals are no longer infectious? Please clarify.

Line 55: sgRNA has not been widely used on clinical diagnostics (I am not aware of its use ever in clinical diagnostics in humans). Please clarify in text.

Line 58: Degradation rate of subgenomic RNA not established in paper cited (Speranza et al.) Limited time points in that study make estimates of differential degradation very difficult. Speranza results between days 1-3 could be explained by lower total expression levels of sgRNA compared to total RNA. The paper also does not provide a calculated half-life for each RNA species.

Line 84. “…thus, no best practices exist to predict and individuals infectiousness.”

There are “best practices” outlined by CDC and are generally based on duration of infectivity from cell culture studies. The authors should consider discussing these or remove the sentence from intro. There are also test-based “best practices” to end isolation with antigen testing. I think the goal of this study is to improve on current “best practices”. The authors should demonstrate this if they can.

Line 256/257: Is “simple model” used here a simple linear regression? Please clarify.

Line 270: Figure 2. Culture logistic. It appears visually that best model is T+DPI+AGE+TG+Aassy+SP +Dose, which seems to have a higher prediction accuracy and MCC than T+DPI+AGE+TG+Aassy+SP +Dose+Cell. Please clarify which is best model.

Line 292

---

## [Decision Letter · Decision Letter 1]

3 Apr 2024

Dear Dr. Lloyd-Smith,

We are pleased to inform you that your manuscript 'PCR data accurately predict infectious virus: a meta-analysis of SARS-CoV-2 in non-human primates' has been provisionally accepted for publication in PLOS Pathogens.

Please consider update the title to reflect methodology advancement of your study that is central to the reported meta-analysis. It could be introduced at the proof stage or earlier; if necessary, please contact Section Editor to approve the change.

Best regards,

Alexander E. Gorbalenya

Section Editor

PLOS Pathogens

Michael Malim

Editor-in-Chief

PLOS Pathogens

orcid.org/0000-0002-7699-2064

Editor's comments:

Please consider update the title with the methodology advancement of your study that is central to the reported meta-analysis. It could be introduced at the proof stage or earlier; if necessary, please contact Section Editor through PP office to approve the change.

Reviewer Comments (if any, and for reference):

Reviewer's Responses to Questions

**Part I - Summary**

Reviewer #2: Summary:

The authors perform a meta-analysis of 24 studies and develop Bayesian statistical models to evaluate the utility of molecular markers as predictors of cell culture infectivity of SARS-CoV-2 from 3 species of experimentally infected non-human primates. Since their first submission, the authors have thoroughly and rigorously addressed both major and minor issues suggested by reviewers of this paper. The manuscript now includes new analyses that provide examples of a clinical application as it relates to duration of isolation. These new analyses add to the relevance of this work. It will be interesting to see how this type of analysis performs on human data in the future.

Overall, this is a very well-done study and makes a nice contribution to the field.

Reviewer #3: The authors addressed all my concerns.

**Part II – Major Issues: Key Experiments Required for Acceptance**

Reviewer #2: None

Reviewer #3: None.

**Part III – Minor Issues: Editorial and Data Presentation Modifications**

Reviewer #2: None

Reviewer #3: None.

PLOS authors have the option to publish the peer review history of their article (what does this mean?). If published, this will include your full peer review and any attached files.

Reviewer #2: No

Reviewer #3: No

---

## [Editor Report · Acceptance letter]

22 Apr 2024

Dear Dr. Lloyd-Smith,

We are delighted to inform you that your manuscript, "Predicting the presence of infectious virus from PCR data: a meta-analysis of SARS-CoV-2 in non-human primates," has been formally accepted for publication in PLOS Pathogens.

Best regards,

Michael Malim

Editor-in-Chief

PLOS Pathogens

orcid.org/0000-0002-7699-2064